# Evolution and regulation of microbial secondary metabolism

**Guillem Santamaria[1,2†], Chen Liao[1†], Chloe Lindberg[1], Yanyan Chen[1], Zhe Wang[3], Kyu Rhee[3], Francisco Rodrigues Pinto[2], Jinyuan Yan[1]\*, Joao B Xavier[1]\***

[1]Program for Computational and Systems Biology, Memorial Sloan Kettering Cancer Center, New York, United States; [2]BioISI – Biosystems & Integrative Sciences Institute, Faculty of Sciences, University of Lisboa, Lisboa, Portugal; [3]Department of Medicine, Weill Cornell Medical College, New York, United States

**Abstract** Microbes have disproportionate impacts on the macroscopic world. This is in part due to their ability to grow to large populations that collectively secrete massive amounts of secondary metabolites and alter their environment. Yet, the conditions favoring secondary metabolism despite the potential costs for primary metabolism remain unclear. Here we investigated the biosurfactants that the bacterium *Pseudomonas aeruginosa* makes and secretes to decrease the surface tension of surrounding liquid. Using a combination of genomics, metabolomics, transcriptomics, and mathematical modeling we show that the ability to make surfactants from glycerol varies inconsistently across the phylogenetic tree; instead, lineages that lost this ability are also worse at reducing the oxidative stress of primary metabolism on glycerol. Experiments with different carbon sources support a link with oxidative stress that explains the inconsistent distribution across the *P. aeruginosa* phylogeny and suggests a general principle: *P. aeruginosa* lineages produce surfactants if they can reduce the oxidative stress produced by primary metabolism and have excess resources, beyond their primary needs, to afford secondary metabolism. These results add a new layer to the regulation of a secondary metabolite unessential for primary metabolism but important to change physical properties of the environments surrounding bacterial populations.

**\*For correspondence:**
jinyuanyan@gmail.com (JY);
xavierj@mskcc.org (JBX)

[†]These authors contributed equally to this work

**Competing interest:** The authors declare that no competing interests exist.

## Editor's evaluation

This important study proposes how *Pseudomonas aeruginosa* is able to produce costly secondary metabolites. By combining a wide variety of experimental and computational approaches and studying 31 isolates in several growth environments, they present highly compelling evidence that secondary metabolites are produced in this species when it is not suffering from oxidative stress linked to its primary metabolism. This paper helps to understand pathogen metabolism, and more generally, how environmental conditions can allow for the evolution of costly collective behavior.

## Introduction

Microorganisms touch practically every human activity, in harmful and beneficial ways: they cause infections (***Morens et al., 2004***) and help us digest food (***Rowland et al., 2018***), make bioterrorism weapons (***Henderson, 1999***) and synthesize life-saving drugs (***Demain and Fang, 2000***), contribute to climate change (***Bardgett et al., 2008***), and clean our wastewater (***Kartal et al., 2010***). The outsize impact of microbes on the macroscopic world that surrounds them comes from their ability to multiply to populations of billions that cooperate to transform chemicals in vast amounts. Many of the chemicals that they release are compounds unessential to growth, development, and division: they

are secondary metabolites. But despite their importance, it remains unclear why microbes invest in secondary metabolites (*Cavalier-Smith, 1992*; *Kell et al., 1995*).

Secondary metabolites enable collective traits like cell-cell signaling (*Dufour and Rao, 2011*), microbial warfare (*Demain and Fang, 2000*), pathogenesis (*Nicas and Iglewski, 1985*), and nutrient scavenging (*Albelda-Berenguer et al., 2019*). These collective traits bring fitness advantages to a population (*Maplestone et al., 1992*) but by diverting precursors, cofactors, and energy away from primary metabolism they can also bring a cost to the individual cell (*Yan et al., 2018*). How did secondary metabolites evolve and how are they maintained by natural selection?

Social evolution theory provides an answer. A secondary metabolite that brings a fitness cost $C$ to the cell and a benefit $B$ to the population can be favored by natural selection if $rB > C$, where $r$ is the relatedness among different genotypes across the population (*West et al., 2006*). This means that secondary metabolite with a high cost can still be evolved if relatedness $r$ is very high, such as in monoclonal populations. But population mixing and mutation, which are frequent in a microbial world (*Andersen et al., 2015*; *Andersen et al., 2018*), mean that microbial populations can have a low relatedness, and a low $r$ limits the maximum cost allowed for secondary metabolites.

Social evolution theory therefore predicts that secondary pathways should have a low cost (*Cavalier-Smith, 1992*). But many secondary products need multiple enzymes for their biosynthesis (*Osbourn, 2010*), and their burden on primary metabolism can be hard to avoid. Microorganisms have sophisticated molecular networks that can sense environmental stimuli and regulate the expression of secondary pathways to lower their cost (*Brakhage, 2013*; *Bibb, 2005*; *Bayram et al., 2016*; *Lind et al., 2015*). Experiments in laboratory culture show, for example, that microbes can regulate the production of secondary metabolites depending on environmental conditions such as the type and quantity of the nutrients provided, especially the carbon source (*Ruiz et al., 2010*), that they can restrict production to specific growth phases, typically the stationary phase (*Malik, 1980*), and that many secondary metabolites are controlled by quorum sensing (*Whiteley et al., 1999*), a type of cell-cell communication that regulates genes in a density-dependent way. Understanding how microbes regulate their secondary metabolites complements the evolutionary explanation by providing a different level of explanation—molecular mechanism—of the same biological phenomenon (*Bateson and Laland, 2013*).

Here we investigate the evolution and regulation of biosurfactants that the bacterium *Pseudomonas aeruginosa* secretes to decrease the surface tension of the surrounding liquid (*Abdel-Mawgoud et al., 2010*). These biosurfactants are secondary metabolites, and they enable spectacular collective behaviors like swarming (*Tremblay et al., 2007*; *Deforet, 2022*), climbing over walls (*Yang et al., 2021*), killing microbial competitors (*Sotirova et al., 2008*), and breaching the epithelial barriers of a host (*Zulianello et al., 2006*). These collective feats, impossible to achieve by a single bacterium, can benefit the population. But a *P. aeruginosa* cell produces 20% or more of its dry weight in biosurfactants (*Guerra-Santos et al., 1984*), which is potentially a huge cost (*Xavier et al., 2011*). A full understanding of *P. aeruginosa* surfactant secretion requires both evolutionary and molecular explanations (*Yan et al., 2019*).

The first step in the biosynthesis of rhamnolipid surfactants is catalyzed by the RhlA enzyme, which takes a metabolic intermediate from fatty acid synthesis, β-hydroxyacyl-acyl carrier protein (β-hydroxyacyl-ACP), and produces 3-(3-hydroxyalkanoyloxy) alkanoic acids (HAAs) (*Zhu and Rock, 2008*). The RhlB and RhlC enzymes each add one unit of rhamnose to the HAAs to produce mono-rhamnolipids and di-rhamnolipids (*Chong and Li, 2017*). The expression of the *rhlAB* operon is regulated by a network that includes at least three quorum sensing signals (*Mukherjee et al., 2018*; *Latifi et al., 1996*) and is conditional to the nutrients in the media, requiring a high carbon-to-nitrogen or carbon-to-iron ratio (*Xavier et al., 2011*; *Boyle et al., 2015*; *Mellbye and Schuster, 2014*). This regulatory strategy—called metabolic prudence—delays the expression of the secondary biosynthetic genes to times when the population is large enough, but also checks if the individual cell has sufficient carbon source to make the product at low or no cost to its fitness (*Xavier et al., 2011*). Metabolic prudence regulates the expression of other secondary metabolites in *P. aeruginosa* besides surfactants (*Mellbye and Schuster, 2014*) and may provide a general evolutionary explanation for secondary metabolites in other microbes as well (*Bruger and Waters, 2015*). Compared to the non-regulated constitutive production, metabolic prudence lowers the $C/B$ ratio and stabilizes biosurfactant synthesis—even when $r$ is low—by preventing cheaters from increasing in frequency (*de Vargas Roditi et al., 2013*).

But this evolutionary explanation alone cannot explain why the surfactant production varies widely among strains, even when they are closely related (*Müller et al., 2011*), and even though the genes encoding the secondary pathway—*rhlA*, *rhlB*, and *rhlC*—are highly conserved (*Germer et al., 2020*). A better understanding of the molecular processes driving surfactant secretion can shed light on the evolution of regulation of this secondary metabolite.

Here we investigate surfactant secretion among diverse *P. aeruginosa* strains that we isolated from infected patients. We start by focusing on the role of glycerol as a carbon source, a relevant nutrient for *P. aeruginosa* infection (*Scoffield and Silo-Suh, 2016*; *Abdel-Mawgoud et al., 2014*) that puts a substantial strain on *Pseudomonas* primary metabolism (*Poblete-Castro et al., 2020*). Then, we investigate other carbon sources to search for a general explanation: we show that the lineages that lost their ability to make the surfactants tend to grow not less but slower compared to lineages that can make surfactants. Using metabolomics and mathematical modeling, we link the inability to reduce oxidative stress produced by primary metabolism with the lack of surfactant secretion. Our results add a new layer to the regulation of surfactant biosynthesis and help explain the inconsistent variation of this secondary metabolite that *P. aeruginosa* secretes to alter the surface tension of its surrounding environment.

## Results

### Surfactant production varies inconsistently across the phylogenetic tree

We investigated the phenotypic variability among clinical isolates in a diverse panel of 31 *P. aeruginosa* strains, including 28 isolates from infected patients (*Yan et al., 2017*) and the 3 types of strains PAO1, PA14, and PA7, all of which have their genomes sequenced. We grew each strain in synthetic media using glycerol as the sole carbon source and ammonium sulfate as the sole nitrogen source, producing a carbon-to-nitrogen molar ratio of 7.0. Then we used the drop collapse assay (*Figure 1A*) to classify each strain as surfactant absent, moderate, or high. Previous work had shown that glycerol enables the biosynthesis and secretion of surfactants in the strain PA14 (*Boyle et al., 2015*), a finding that we confirmed here with the wild-type PA14 and its isogenic mutants in the rhamnolipid synthesis. As expected, the wild type caused drop collapse indicating surfactant secretion, the Δ*rhlA* mutant did not, indicating loss of surfactants, and the complementation of the Δ*rhlA* with the L-arabinose inducible $P_{BAD}rhlAB$ restored the drop collapse but only when L-arabinose was added, as expected (*Figure 1B*). Among the clinical isolates, however, there was a wide variability of surfactant phenotypes. The strongest producers (n=17) showed phenotypes similar to the wild-type PA14, while non-producers (n=8) showed no surfactant secretion similar to the Δ*rhlA* mutant; moderate producers (n=6) showed a phenotype in between. There was no obvious relation between the body site infected in the patient and the surfactant phenotype (*Figure 1C*). There was also no coherent distribution with the isolates' phylogeny, which we built using the sequence variation in core genes, also supported by a poor correlation in Moran's I test of p=0.14. Ancestral reconstruction of the surfactant secretion trait supports that the ability to make surfactants from glycerol was present in the common ancestor of all strains (*Figure 1—figure supplement 2*). Importantly, the genes *rhlA*, *rhlB*, and *rhlC*, the genes encoding the secondary biosynthesis pathway, were present in all 31 isolates, and their sequence was 100% conserved. This indicates that the absence of surfactant secretion was not due to a loss of the biosynthetic pathway. Instead, we hypothesized that the phenotypic variation was due to the life histories of the strains: some lineages may have lost their ability to rewire their regulatory network to overcome the specific selective pressures that they had faced during their evolution.

We further investigated whether variation elsewhere in the genome—particularly in the accessory genome (the set of genes missing in at least one strain)—could explain the phenotype differences. Among the 8290 accessory genes, 354 are only absent in those non-producers but present in all mild and strong producers (*Supplementary file 1*). In a principal component analysis (PCA) of the presence-absence profiles, there was no visible grouping of isolates by their surfactant phenotype (*Figure 1—figure supplement 2*), which is consistent with the lack of association with the phylogenetic tree built from the core genome.

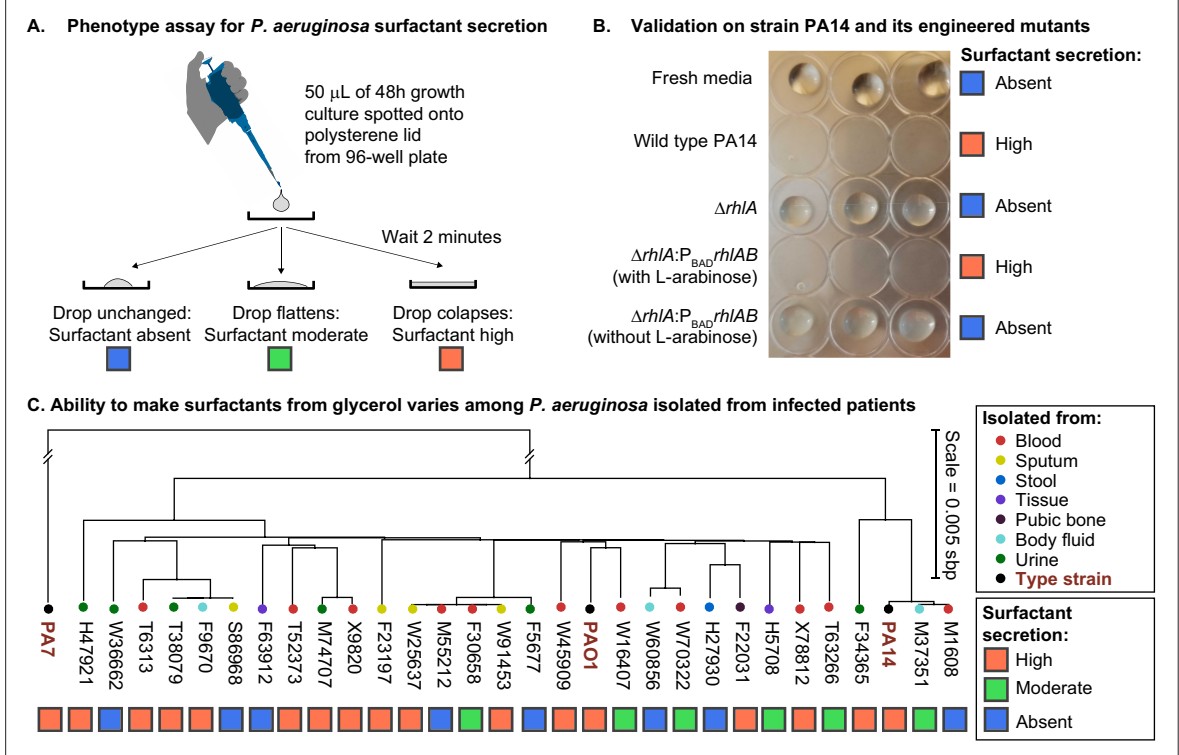

**Figure 1.** The ability to make surfactants from glycerol and reduce the surface tension of its liquid medium varies widely among clinical isolates of *Pseudomonas aeruginosa*, is uncorrelated with phylogeny, and is not associated with the tissue of origin. (**A**) The drop collapse assay assesses the ability of *P. aeruginosa* to reduce the surface tension of its media by looking at the shape of a drop placed onto polystyrene. (**B**) The phenotypic assay in the wild-type PA14 (which can make surfactants from glycerol) is compared with the PA14 Δ*rhlA* isogenic mutant (which does not produce the rhamnolipid biosurfactants because it lacks a key biosynthetic gene). We also included the complementation mutant PA14 Δ*rhlA* P_BAD*rhlA* with and without the inducer of the P_BAD promoter, L-arabinose. As expected, the ability to reduce surface tension is only present in the PA14 wild type and in the induced P_BAD mutant. The Δ*rhlA* and the P_BAD mutant in the absence of L-arabinose show high surface tensions like the fresh media. (**C**) A phylogenetic tree built from the core genome of 28 clinical isolates obtained from patients with cancer at Memorial Sloan Kettering Cancer Center (***Lind et al., 2015***; ***Cai et al., 2017***) and the three type strains PAO1, PA14, and PA7. The tissue where each isolate was originally obtained from is indicated by the colored circles. The surfactant phenotype was assessed for all strains using the drop collapse assay after growth in glycerol minimal medium. The phenotype shows no obvious pattern with phylogeny or tissue of origin.

The online version of this article includes the following figure supplement(s) for figure 1:

**Figure supplement 1.** Phylogenetic ancestor state reconstruction of surfactant secretion from glycerol minimal media.

**Figure supplement 2.** A principal component analysis (PCA) of the matrix of presence/absence of genes in each strain.

## Surfactant secretion lost in strains with slower exponential growth

To search for a general explanation, we conducted analyses that showed producers have faster primary metabolisms in the same media of drop collapse assay. This finding was unexpected because, if anything, secondary metabolism should divert resources from primary metabolism (***Yan et al., 2018***): strains that produce surfactants would grow slower, not faster. We started by noting differences among the growth curves of the 31 strains, as assessed by the dynamics of the population optical density at 600 nm ($OD_{600}$). The maxima that the population densities reached showed no obvious difference in producers vs non-producers, but the cultures did vary widely in the shape of their lag phase and the exponential phase (***Figure 2A***). The shape of the growth curve, as a whole, was not predictive of surfactant secretion because hierarchical clustering using the entire time series failed to separate producers and non-producers (***Figure 2—figure supplement 1***). Instead, the slope of the exponential growth phase was steeper in producers. This was also shown using non-negative matrix factorization (***Lee and Seung, 1999***), which could decompose each growth curve as a weighted sum of three basis functions (i.e. features). This analysis showed that—although the growth curves of the producers (orange lines; including mild and strong producers) and non-producers (blue lines)

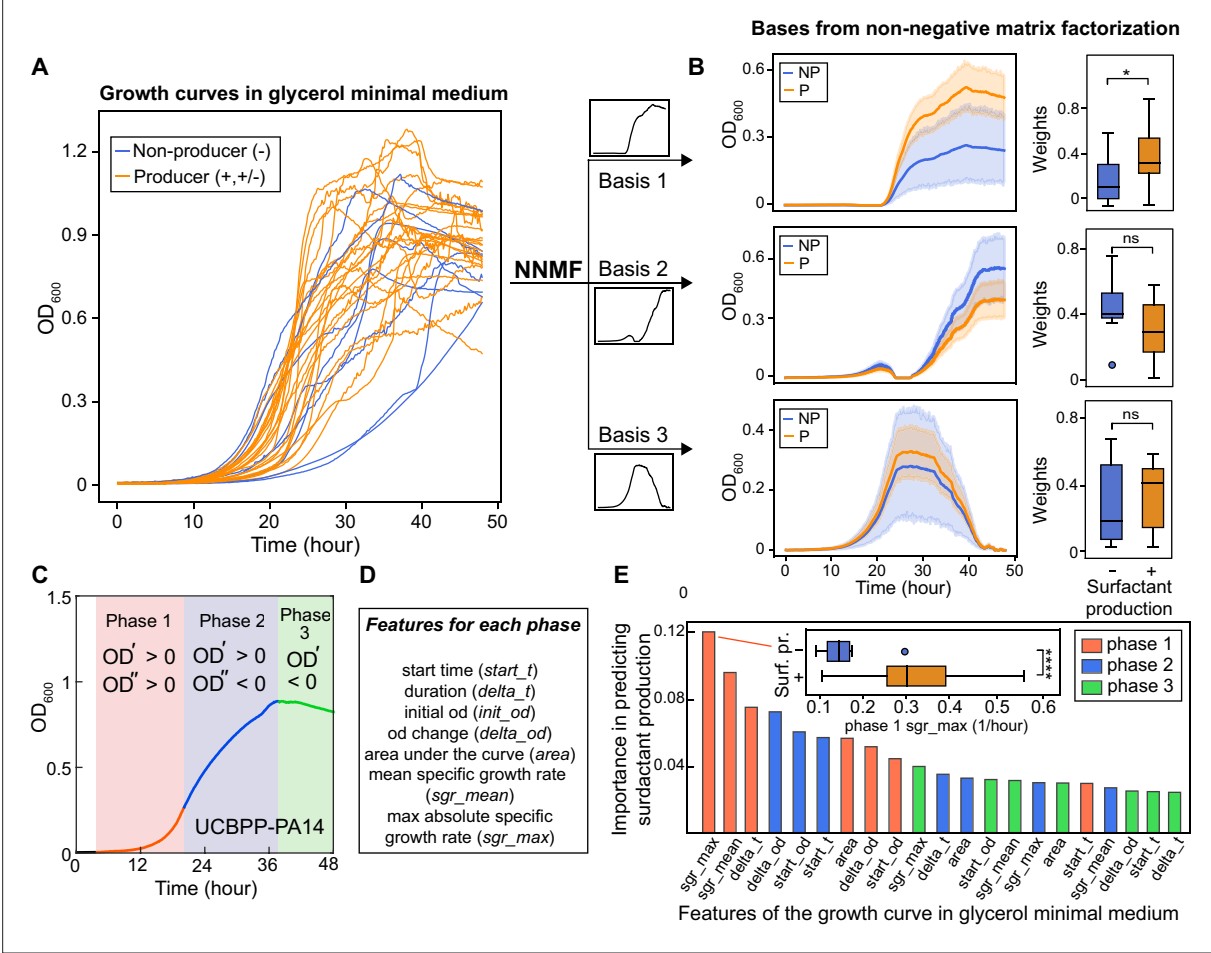

**Figure 2.** The shape of the growth curve in glycerol minimal media distinguishes surfactant producers from non-producers. (**A**) Growth curves obtained in glycerol minimal media for producers (high and moderate, in orange) and non-producers (in blue). (**B**) We used two types of analysis: first, non-negative matrix factorization (NNMF) was used to decompose the growth curves into three additive basis functions (features). Each growth curve can be approximately represented by the weighted sum of these functions. The components 1 and 2 (basis function multiplied by weights; left panels) and weights (right panels) from NNMF of surfactant producers differ from non-producers. The shaded areas represent 95% bootstrap confidence intervals of the mean. (**C**) Second, we used supervised feature selection by random forest classifier, first divides each growth curve (excluding the initial lag phase) into three phases. (**D**) Each phase is described by seven quantitative features. (**E**) The random forest analysis quantifies the importance of each feature in distinguishing producers from non-producers. Inset: boxplot of maximum specific growth rate of phase I grouped by surfactant production. Welch's t-test was used in (**D**) and (**G**) for significance testing. ****, p-value ≤0.0001; *, p-value ≤0.05; ns, p-value >0.05.

The online version of this article includes the following figure supplement(s) for figure 2:

**Figure supplement 1.** Hierarchical clustering of growth curves of *Pseudomonas aeruginosa* clinical isolates and three type strains PA14, PAO1, and PA7 in glycerol minimal medium.

**Figure supplement 2.** Growth curve of *Pseudomonas aeruginosa* strains in glycerol minimal medium.

largely overlapped (*Figure 2A*)—producers had higher weights for base 1, which agrees with a faster growth rate in the exponential phase (*Figure 2B*). This was also confirmed by dividing each growth curve into three phases (*Figure 2C* and *Figure 2—figure supplement 2*) and defining seven quantitative features to characterize each growth phase (*Figure 2D* and *Supplementary file 2*). Random forest classification revealed that the top two features associated with surfactant production were the maximum and the average specific growth rates in phase I, the exponential growth phase (*Figure 2E*). Features of phase III (the decay phase), such as the starting value of $OD_{600}$ in phase III which represents the final biomass produced by different strains in phase II, were among the least significant.

These analyses indicate that the ability to make surfactants, which typically peak when growth slows down from exponential (*Latifi et al., 1996*; *Boyle et al., 2015*; *Mellbye and Schuster, 2014*),

is associated with the speed of the exponential phases, and hence, an efficient primary metabolism on glycerol. This suggests that the non-producer lineages had particular life histories that are selected for specific metabolic adaptations, and that those adaptations had the collateral damage of impacting their ability to grow on glycerol as a sole carbon source.

## Metabolomics shows differences in intracellular metabolomes associated with loss of surfactant biosynthesis

We then searched the composition of the intracellular metabolome for differences that distinguish producer and non-producer lineages. We extracted the metabolites from all our strains except for two non-producers (M55212 and F23197), which grew too slowly during the transition between phase I and phase II when rhamnolipid production begins (Materials and methods). Liquid chromatography-mass spectrometry (LC-MS) revealed 67 metabolites whose identities are known, and their abundances varied significantly across the tested strains (*Figure 3—figure supplement 1A–B*). Hierarchical clustering and PCA (*Figure 3—figure supplement 2A–B*) showed a reasonable separation between

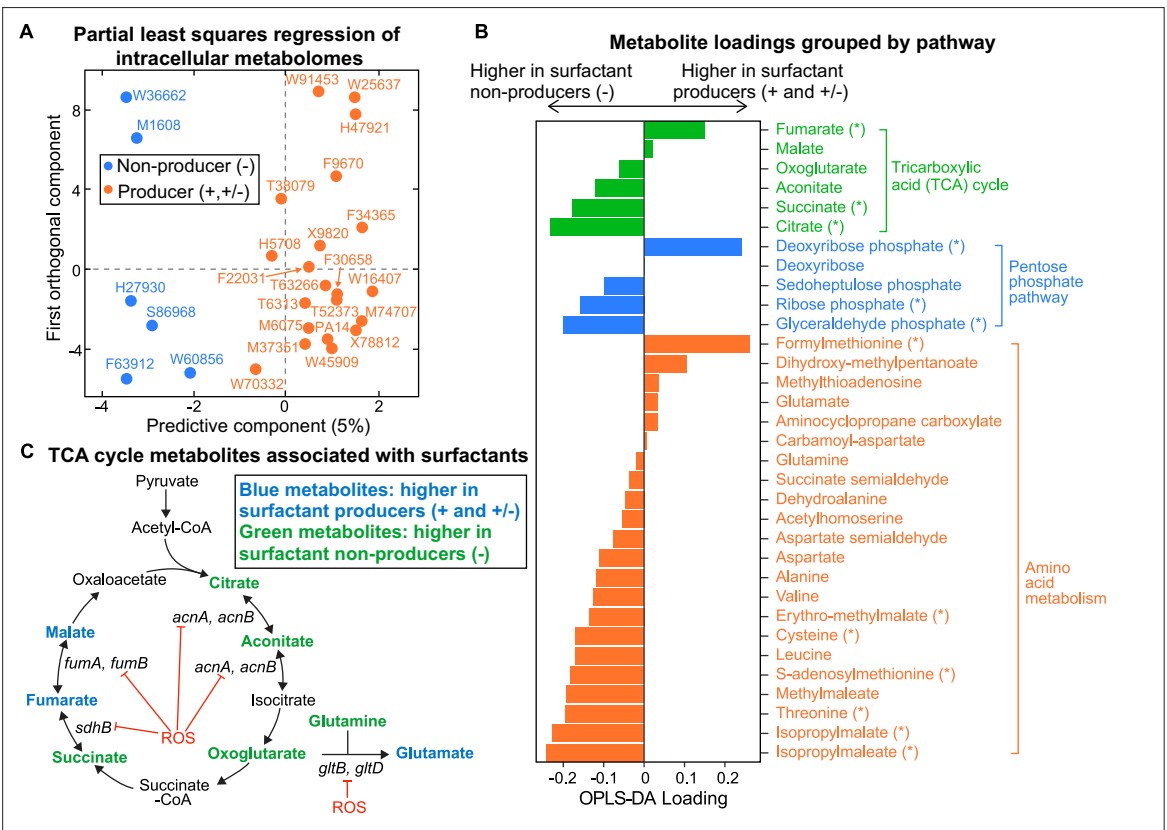

**Figure 3.** Intracellular metabolites measured in glycerol minimal media differ between surfactant producers and non-producers. (**A**) OPLS-DA (orthogonal partial least squares discriminant analysis) was used to compare the intracellular metabolomes of producers (high and moderate) and non-producers. The predictive component is high in producers and low in non-producers. (**B**) OPLS-DA loading values for the predictive component of a selected number of metabolites. Metabolites with asterisk (*) are significantly different between producers and non-producers (adjusted p-value of a Mann-Whitney test <0.05). (**C**) Differential abundance of metabolites involved in reactions catalyzed by some Fe-S-containing enzymes whose activities are inhibited by reactive oxygen species (ROS). Abbreviations: *acnA*: aconitate hydratase A, *acnB*: aconitate hydratase B, *sdhB*: succinate dehydrogenase subunit, *fumA*: fumarase A; *fumB*: fumarase B; *gltB*: glutamate synthase subunit; *gltD*: glutamate synthase subunit.

The online version of this article includes the following figure supplement(s) for figure 3:

**Figure supplement 1.** Metabolites identified in this study were examined and imputed to represent the profile of all 31 strains for further analysis.

**Figure supplement 2.** Variance and clustering analysis of metabolomics across *Pseudomonas aeruginosa* strains.

**Figure supplement 3.** The loading values of all predictive metabolites of the orthogonal partial least squares discriminant analysis model.

**Figure supplement 4.** Volcano plot of metabolomics data between wildtype (WT) *Pseudomonas aeruginosa* PA14 strain and its Δ*rhlA* mutant grown in glycerol minimal medium (replotted with permission from *Bayram et al., 2016*).

producers and non-producers. Fitting the data using the orthogonal projections to latent structures-discriminant analysis (OPLS-DA) (*Cloarec et al., 2005*) confirmed that there is a significant association between the broad metabolome and surfactant secretion (*Figure 3A*, $R^2$=0.82, $Q^2$=0.66, p-value = 5e−4). A Mann-Whitney U test revealed the 15 metabolites whose abundances differed most significantly between producers and non-producers. Then, we used those 15 metabolites to identify pathways perturbed in non-producers using the FELLA algorithm for pathway enrichment analysis (*Picart-Armada et al., 2018*; *Figure 3—figure supplement 3*, *Supplementary file 3*). The most perturbed pathways were the TCA cycle and amino acid metabolism (*Figure 3B*). Three out of six metabolites in the TCA cycle—fumarate, succinate, and citrate—were significantly changed: fumarate and malate had positive loadings, which indicated higher levels in producers. Citrate, succinate, and to a lesser extent cis-aconitate and alpha-ketoglutarate had negative loadings, which indicated higher levels in non-producers (*Figure 3B*). Pyruvate remained relatively constant across all the strains (*Figure 3—figure supplement 1A*), implying that the differential responses in the TCA cycle were independent from the changes in its upstream central carbon metabolism.

Besides the TCA cycle metabolites, most of the annotated compounds in the metabolism of branched chain amino acids (leucine/isoleucine, valine) and sulfur-containing amino acids (cysteine/methionine) were more abundant in non-producers than producers (*Figure 3—figure supplement 3*). A notable exception was formylmethionine (fMet), which was lower in non-producers. To place this observation in context, we reanalyzed previous metabolomics data (*Boyle et al., 2017*) from an experiment where we had compared PA14 and its Δ*rhlA* mutant. Interestingly, the Δ*rhlA* mutant also had lower levels of fMet compared with the wild type (*Figure 3—figure supplement 4*). The Δ*rhlA* mutant grows just as fast in glycerol as the wild type (*van Ditmarsch and Xavier, 2011*), which means that the link between lower fMet and lack of surfactant secretion is not simply due to a growth defect.

## Theory formalizes the link between oxidative stress, slower growth, and lack of surfactants

Several theoretical arguments support that non-producers might have slower primary metabolisms because they suffer from higher oxidative stress than the producers. First, the major differences include TCA intermediates (*Figure 3B*). The TCA cycle harbors five enzymes with Fe-S clusters, aconitase A, aconitase B, succinate dehydrogenase (SDH), fumarase A, and fumarase B (*Py and Barras, 2010*), which make them especially vulnerable to oxidative stress (*Figure 3C*). If non-producers are less successful at removing the oxidative stress from growth in glycerol, then the stress accumulated could reduce flux through the TCA cycle, which would slow down growth. The accumulation of succinate and depletion of fumarate in non-producers could be due to a reduced activity of SDH under oxidative stress: SDH is a membrane-bound dehydrogenase linked to the respiratory chain—a major site of ROS production in the cell—and also a member of the TCA cycle that catalyzes the oxidation of succinate into fumarate (*Hederstedt and Rutberg, 1981*). Since SDH contains [2Fe-2S], [3Fe-4S], and [4Fe-4S] clusters (*Ayala-Castro et al., 2008*), ROS that damages Fe-S clusters could decrease SDH activity in vivo.

Second, we saw that non-producers also included intermediate metabolites in amino acid biosynthetic pathways which are also substrates of Fe-S containing enzymes (*Figure 3B*). For example, the glutamate synthetase has a large chain (encoded by *gltB*) and a small chain (encoded by *gltD*), and both subunits contain Fe-S clusters that catalyze the production of glutamate from oxoglutarate and glutamine (*Curti et al., 1996*). Non-producers had slightly higher oxoglutarate and glutamine as well as lower glutamate than producers.

Finally, we used a whole-genome reconstruction to model the intracellular metabolic fluxes during growth in the glycerol medium (*Nogales et al., 2017*; *Nogales et al., 2020*). The rhamnolipids are produced when carbon is in excess (*Boyle et al., 2015*; *Mellbye and Schuster, 2014*) and, in the simulations, rhamnolipids secretion occurred when C:N flux ratio >6.3 (*Figure 4—figure supplement 1*). We then set C:N to 10.0, which exceeds the minimum threshold, and we simulated various levels of redox stress level by changing the flux through three redox molecules—NADH (reduced nicotinamide adenine dinucleotide), NADPH (reduced nicotinamide adenine dinucleotide phosphate), and GSH (reduced glutathione)—responsible for the bulk of cellular electron transfer and the main sources of reactive oxygen species (ROS) (*Xiao and Loscalzo, 2020*). Although redox stress is ultimately caused by high ROS levels, not fluxes, and ROS was not an explicit variable in our model, we assumed that

ROS could perturb intracellular metabolic fluxes, particularly the fluxes of redox molecules, and we then asked how cell growth and surfactant synthesis responded to these ROS-mediated indirect perturbations.

For all three redox molecules, we found that the maximum growth rate was maintained at an intermediate flux range (redox homeostasis; *Figure 4A–C*, upper panels, white area). The growth rate decreased gradually as the flux of redox molecules changed in either direction away from this homeostatic interval (gray shading). The reason for the gradual decrease in growth rate is that deficiency or excess of these redox molecules which participate in the redox control of a great variety of biological processes disturbs the primary metabolism for growth. Importantly, before the growth starts slowing down due to an excess in redox species there was an abrupt shutdown in the overflow on carbon into many central carbon metabolites and—importantly— in the secretion of the HAAs, mono- and di-rhamnolipids that makeup the surfactants (*Figure 4A–C*, lower panels). The shutdown of overflow reactions under excess of redox molecules mimics the metabolic state in the non-producer strains who lose metabolic plasticity to overflow metabolites without compromising growth (*Vemuri et al., 2006*). When we forced these secretion fluxes to zero—to simulate a ΔrhlA mutant—the simulation did not predict a growth benefit. This result agrees with data showing that the ΔrhlA mutant has no growth benefit in glycerol media compared with wild type (*van Ditmarsch and Xavier, 2011*).

This also suggests that regaining surfactant synthesis should not restore fast growth in glycerol to non-producers, which agrees with the idea that loss of surfactant secretion is a consequence—not a cause—of the slower growth in non-producers. Notably, most of the overflowed metabolites are carbon-rich molecules, including HAA, mono-, and di-rhamnolipid (*Xavier, 2011*). Collectively, these theoretical arguments support a link between the ability to make surfactants and oxidative stress: fast growth and surfactant secretion are both impacted by the accumulation of ROS produced by primary metabolism.

## Non-producers are worse at reducing oxidative stress

Our model suggested that non-producers suffer from higher oxidative stress. *Pseudomonas* have a variety of antioxidant enzymes including catalases, glutathione reductases, NADH peroxidases, and cysteine-based peroxidases (*Mishra and Imlay, 2012*), which in PA14 and PAO1 were only made when the growth rate maintained its maximum value under excess carbon (*Xavier et al., 2011*; *Boyle et al., 2015*; *Mellbye and Schuster, 2014*). Isogenic mutants lacking some of those could make surfactants (*Vinckx et al., 2010*). If the slower growth in the clinical non-producer strains is due to impairments in some of those mechanisms to reduce oxidative stress, then this could affect surfactant biosynthesis.

To test this hypothesis, we compared the dynamics of hydrogen peroxide ($H_2O_2$) during growth in glycerol (*Figure 4D and E*). $H_2O_2$ is a representative ROS that can diffuse freely between cells and the environment; we can access its dynamics by measuring the fluorescence by the Amplex assay, where the difference between cell cultures and baseline reveals each strain's net ability to reduce $H_2O_2$ (*Zhao et al., 2012*). All strains except for M1608, one of the non-producers, quenched the signal over the first 18 hr (*Figure 4E*). Consistent with our hypothesis, the non-producers (blue lines) removed less $H_2O_2$ than moderate producers (green lines) and strong producers (red lines). To factor out the possibility that the non-producers performed worse due to lower cell density, we calculated the $H_2O_2$ removal rate per cell. We observed a similar relative trend of $H_2O_2$ degradation between non-producers and producers (*Figure 4F*): a linear mixed-effect model quantified the difference that mild and strong producers have in per-cell $H_2O_2$ removal rates, which was in both cases significantly higher than in non-producers ($p<0.001$; *Figure 4G*). This is consistent with the notion that non-producers suffer more from oxidative species, which can be caused by high production or slow degradation.

To identify the genes that would explain the varying tolerance to oxidative stresses, we performed a Spearman correlation between gene presence or absence and the time averaged $H_2O_2$ removal rate per cell. No gene passed the significance threshold ($p<0.05$) after Benjamini-Hochberg correction for multiple hypothesis testing. It is possible that oxidative stress tolerance is a system-level phenotype that depends on the interactions among many genes in multiple pathways. Alternatively, the phenotype can be determined by a small number of, but different genes for different strains. For example, the worst strain in this assay, M1608, lacks *katE* (*Supplementary file 1*), a catalase that degrades $H_2O_2$ (*Mulvey et al., 1988*), and another non-producer, F5677, lacks the redox sensor *soxR* (*Hidalgo et al., 1998*).

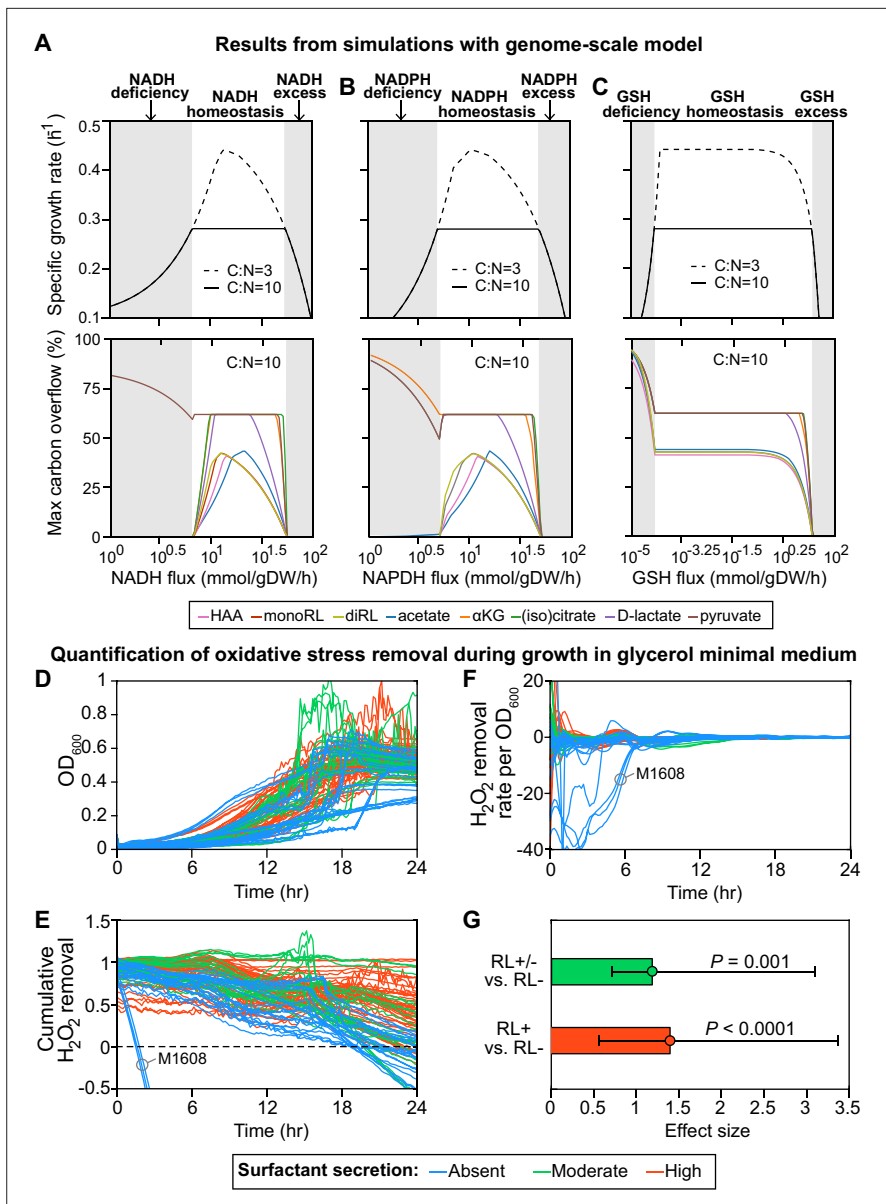

**Figure 4.** Computer simulations using flux-balance analysis and experiments with $H_2O_2$ indicate that the ability to produce surfactants in glycerol media depends on a strain's ability to reduce oxidative stress. (**A-C**) Computer model: the simulations vary the redox balance of *Pseudomonas* metabolism by altering the fluxes of (**A**) NADH (reduced nicotinamide adenine dinucleotide), (**B**) NADPH (reduced nicotinamide adenine dinucleotide phosphate), and (**C**) GSH (reduced glutathione). The upper panels show the predicted maximum growth rates, and lower panels are predicted maximum byproduct secretion fluxes. C:N indicates the carbon-to-nitrogen influx ratio provided by the culture medium. C:N=3 and C:N=10 represent carbon- and nitrogen-limiting conditions, respectively. Abbreviations: HAA: 3-(3-hydroxyalkanoyloxy) alkanoate; monoRL: monorhamnolipid; diRL: dirhamnolipid; aKG: alpha-ketoglutarate. (**D–G**). Experimental comparison of the ability to remove hydrogen peroxide ($H_2O_2$) among strong surfactant producers (+), weak producers (±), and non-producers (−). (**D**) Population density ($OD_{600}$). (**E**) The total amount of hydrogen peroxide removed from the environment. Negative values indicate net cellular production of hydrogen peroxide released to the environment. (**F**) The specific hydrogen peroxide removal rate. In both (**E**) and (**F**), each trajectory of $H_2O_2$ fluorescence intensity was normalized to the averaged trajectory of the wild-type PA14 strain. (**G**) Effect size of surfactant production as a predictor of $H_2O_2$ removal rate per unit of $OD_{600}$ determined by a linear mixed-effect model shows that producers can reduce oxidative stress better than non-producers.

The online version of this article includes the following figure supplement(s) for figure 4:

*Figure 4 continued on next page*

**Figure supplement 1.** Theoretical estimation of threshold carbon (glycerol):nitrogen (ammonium) ratio above which carbon is in excess in the sense that carbon release through rhamnolipids and central carbon metabolites does not compromise biomass production.

## RNAseq reveals oxidative stress response in non-producers

To investigate transcriptomic differences associated with the surfactant phenotype, we selected 11 representative strains (6 strong producers, 3 mild producers, and 2 non-producers), in proportion to the number in each category. We then assessed their transcriptomics in Phase II, which is typically when producers start their surfactant biosynthesis (*van Ditmarsch and Xavier, 2011*). A PCA of the gene expression profiles could not cluster the strains according to their phenotypes (*Figure 5—figure supplement 1*). We then turned to RLQ analysis (R, L, and Q are explained in *Figure 5A*) to find both genes (*Figure 5B*) and pathways (*Figure 5C*) associated with rhamnolipid production.

As expected, the genes in the biosynthesis of rhamnose (*rmlA, rmlB, rmlC, rmlD*), rhamnolipid biosynthesis (*rhlA, rhlB, rhlC, rhlR, rhlI*), and quorum sensing (*lasA, lasB, lasI, pqsABCDEH*) were significantly less expressed in the non-producers (*Figure 5B*, *Figure 5—figure supplement 2*). Interestingly, in vitro data shows that the expression of genes from the above list, including *lasI, lasR, rhlI, rhlR, pqsA, pqsR*, in *P. aeruginosa* PAO1, can be significantly reduced by quorum sensing quencher and oxidative stress by exogenous $H_2O_2$ (*Mohamed et al., 2020*), supporting that the loss of the phenotype is linked to oxidative stress. More importantly, the *zwf* gene encoding glucose 6-phosphate dehydrogenase (G6PDH) was expressed higher in non-producers (*Figure 5—figure supplement 2*). G6PDH, a major source of NADPH that contributes to antioxidant defenses by reducing oxidized glutathione, was expressed higher, supporting our hypothesis that non-producers experience higher oxidative stress. Other genes, including the peroxide-sensing transcriptional regulator (*oxyR*), catalases (*katA, katB*), alkyl hydroperoxide reductases (*ahpC, ahpF*), glutaredoxin (*grx*), glutathione reductase (*gor*), and thioredoxins (*trxA, trxB1, trxB2*) were not highly expressed in the producers (*Figure 5—figure supplement 2*). We speculate that the stress levels may occur locally within the cells (e.g. within SDH) so that the oxidative stress response is also local. Another possibility is that the antioxidant defense may be regulated at the metabolic level. A notable example is biosynthesis of phenazine, whose genes (e.g.

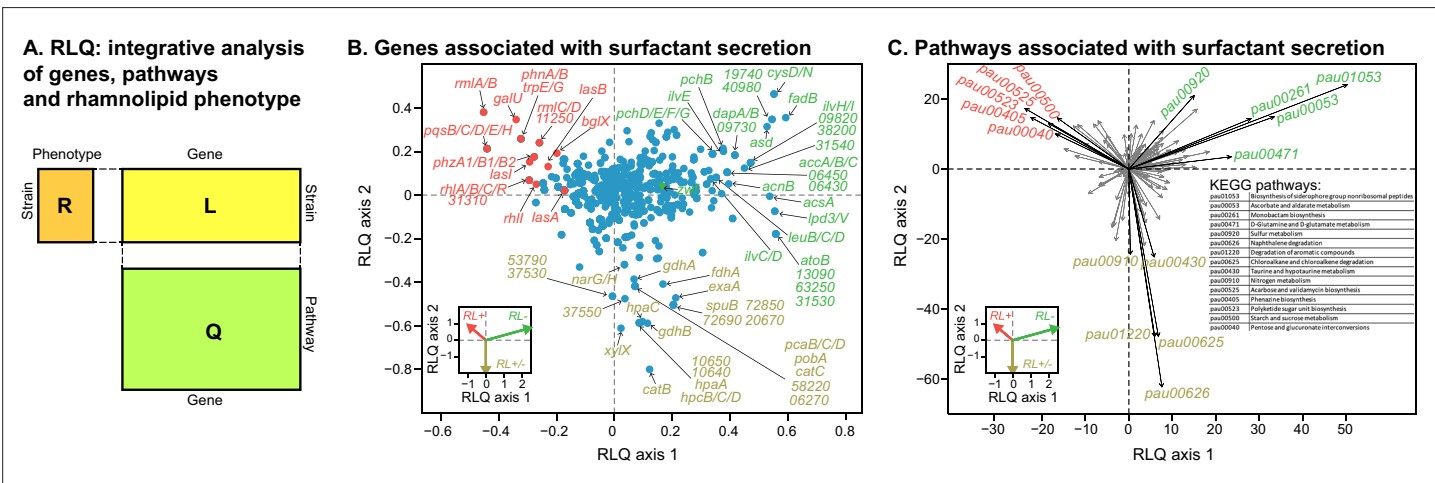

**Figure 5.** Transcriptomic data, analyzed using the RLQ analysis, reveals genes and metabolic pathways with expression associated with the ability to produce surfactants in glycerol minimal media. (**A**) RLQ provides a way to analyze simultaneously associations from three tables. The results reveal not only the genes (**B**) but also the pathways (**C**) whose expression is associated with the rhamnolipid phenotype. The insets in both (**B**) and (**C**) represent the surfactant secretion phenotype (assessed in glycerol) in the two principal RLQ axes. Genes and pathways align with the direction of each phenotypic category: in panel (**C**), the top five pathways ('pau' are KEGG pathway IDs) associated with each phenotype are highlighted in black. The five-digit number in panel (**C**) represents the PA14 locus ID.

The online version of this article includes the following figure supplement(s) for figure 5:

**Figure supplement 1.** Principal component analysis (PCA) plot of RNA expression colored by surfact production phenotypes in glycerol minimal media.

**Figure supplement 2.** Comparisons of expression for selected genes across strains with different abilities of surfactant biosynthesis from glycerol.

*phzA1/B1/B2*) are highly expressed in the producers but not in the non-producers. It was previously shown that the phenazine pyocyanin plays a role in the reduction of oxidative stress as it can restore growth of the $\Delta oxyR$ mutant in rich medium (LB) and undergoes $H_2O_2$ oxidation (*Vinckx et al., 2010*).

The RLQ analysis also revealed the top five pathways enriched specifically in each phenotype group: strong producers were enriched in acarbose and validamycin biosynthesis (pau00525), phenazine biosynthesis (pau00405), polyketide sugar unit biosynthesis (pau00523), starch and sucrose metabolism (pau00500), and pentose and glucuronate interconversions (pau00040) (*Supplementary file 4*). By contrast, non-producers were enriched in the biosynthesis of siderophore group nonribosomal peptides (pau01053), ascorbate and aldarate metabolism (pau00053), monobactam biosynthesis (pau00261), D-glutamine and D-glutamate metabolism (pau00471), and sulfur metabolism (pau00920).

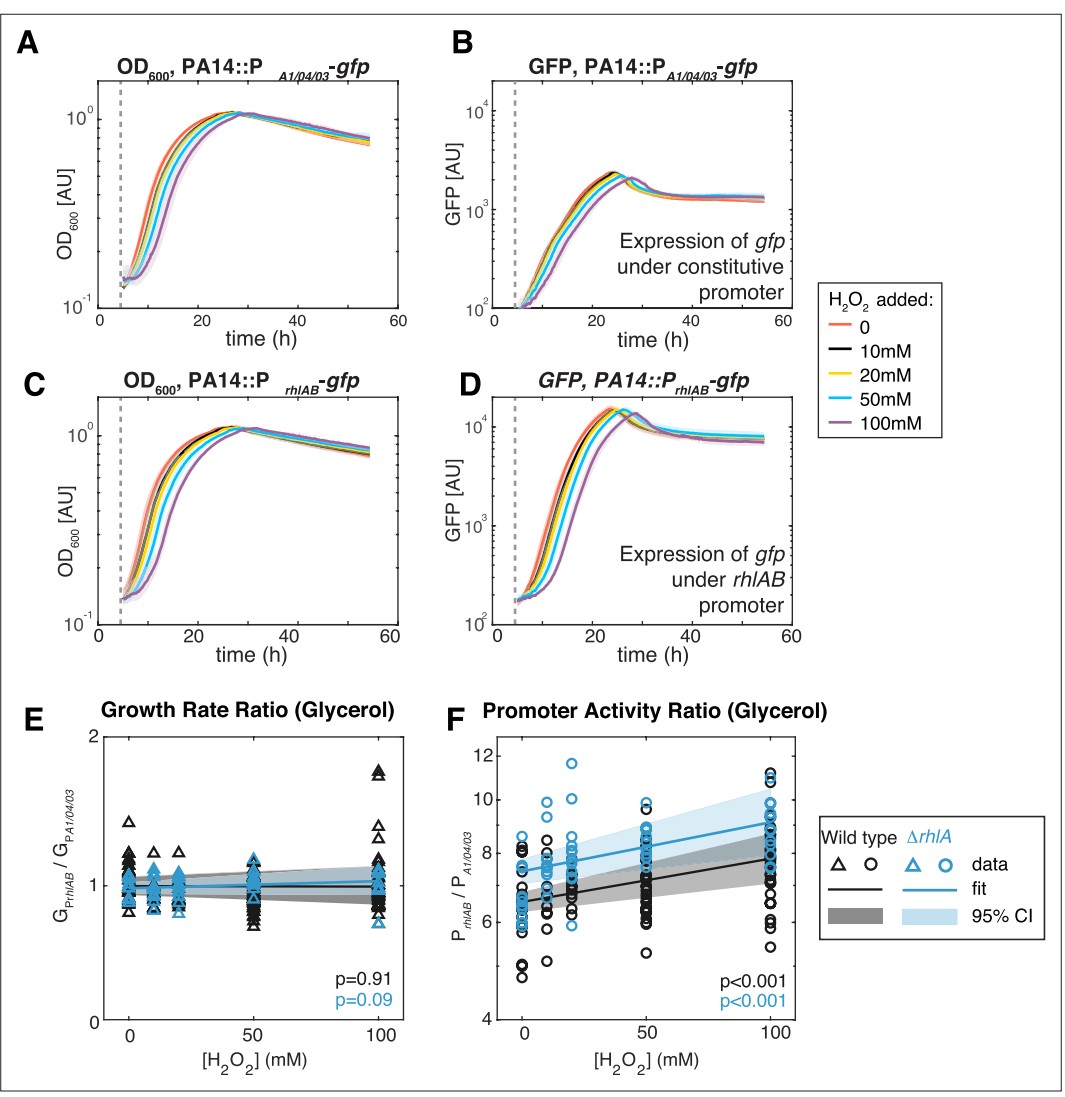

**Figure 6.** Extrinsic oxidative stress by adding $H_2O_2$ increases $P_{rhlAB}$ activity without impacting cell growth in the stain PA14 in glycerol medium. (**A–D**) Growth (**A, C**) and *gfp* expression (**B, D**) of PA14::$P_{A1/04/03}$-GFP (constitutive promoter) and PA14::$P_{rhlAB}$-GFP (*rhlAB* promoter) with increasing levels of $H_2O_2$ added to the media at the beginning of exponential growth (gray dash line). The shaded area represents the range of all replicates from the same experiment. (**E, F**) The ratio of growth rate (**E**) and promoter activity (**F**) of PA14::$P_{rhlAB}$-GFP to PA14::$P_{A1/04/03}$-GFP in phase II under different levels of $H_2O_2$ in glycerol minimal media. The p-value quantifies the significance of the coefficient (slope) obtained from linear fitting.

## Exogenous oxidative stress impacts expression of surfactant biosynthetic genes

So far, we have shown data supporting that surfactant producers can reduce ROS better than non-producers. This explains why strains that grow slower in glycerol also tend to produce less surfactants from that carbon source, likely due to the impact of ROS on growth. It is even possible that the ability to direct metabolic fluxes to surfactant secretion is part of a multifarious response to oxidative stress. Previous work in an engineered strain of *P. putida* showed that inducing surfactant synthesis reduces production of the final electron acceptor $CO_2$, suggesting that directing metabolic flux to surfactant secretion could indeed serve as an alternative electron sink (*Tiso et al., 2016*).

To investigate the link with oxidative stress further, we measured the expression of the rhamnolipid synthesis operon *rhlAB* to extrinsic oxidative stress using PA14 genetically engineered with a reporter fusion ($P_{rhlAB}gfp$). This strain allows tracking the expression of *rhlAB* through a dynamic readout of green fluorescence (*Figure 6*). A PA14 carrying *gfp* under a constitutive promoter, $P_{A1/04/03}gfp$ (*Lambertsen et al., 2004*), was used in parallel to account for growth effects on the *gfp* dynamics. We then added $H_2O_2$ at different doses always at the time when the culture started entering the exponential phase of growth in glycerol. In both PA14::$P_{rhlAB}gfp$ and PA14::$P_{A1/04/03}gfp$, the start of the exponential phase was increasingly delayed as the concentration of $H_2O_2$ was larger (*Figure 6A and C*), but once the exponential growth started there was no difference in the growth rate under different $H_2O_2$ concentrations (*Figure 6E*). This indicates that PA14 can restore normal operation of its primary metabolism after removal of the extrinsic stress.

The apparent removal of $H_2O_2$ during the lag phase may be achieved by metabolic rewiring that affects rhamnolipid production in the late exponential phase (phase II). Similar to growth dynamics, the promoter activity of both mutants was also delayed, and the time delay is longer at higher $H_2O_2$ concentration (*Figure 6B and D*). We determined that the ratio of the median $P_{rhlAB}$ expression relative to the median $P_{A1/04/03}$ increased linearly with the $H_2O_2$ concentration (*Figure 6F*, orange). Since $P_{A1/04/03}$ measures the growth dependence of promoter activity (e.g. changes in number of RNA polymerases and ribosomes), the increased ratio showed that extrinsic redox stress can induce *rhlAB* expression via a growth-independent mechanism that likely involves transcriptional activators.

We next compared the above results obtained using PA14 wild-type strain and the Δ*rhlA* mutant. The *gfp* dynamics showed that in the absence of $H_2O_2$ the Δ*rhlA* strain still attempts to express *rhlAB* at a similar level as the wild type (p=0.3). We observed no decrease in the growth rate with increasing $H_2O_2$ concentration (*Figure 6E*), which indicates that the lack of surfactant biosynthesis had no detrimental impacts on the primary metabolism. Similar to the PA14 wild-type strain, $H_2O_2$ increased the expression of the *rhlAB* promoter relative to the constitutive promoter in the Δ*rhlA* mutant (*Figure 6F*, blue), but the baseline promoter activity increased significantly compared to the wild type (1.14×, p<0.01). Taken together, these results indicate that the genes for surfactant biosynthesis can be induced by extrinsic redox stress, and that the induction is stronger if surfactant synthesis fails to start.

## Mathematical model predicts impact of carbon sources on surfactant production

All analyses so far were done in glycerol, a carbon source relevant for *P. aeruginosa* rhamnolipid production in bioremediation (*Zhao et al., 2021*) and clinical settings (*Scoffield and Silo-Suh, 2016*; *Abdel-Mawgoud et al., 2014*). But could different environments lead strains that are non-producers in glycerol to produce surfactants, and vice-versa? To address this question, we developed a predictive model from data and then sought out to test its predictions experimentally.

The model is a function $f$ that takes as input a given carbon source $m_i$ and predicts whether a strain produces surfactants from that source. To parameterize the model we obtained a dataset using the BIOLOG phenotype microarrays PM1 and PM2a; these two arrays together contain 190 distinct carbon sources in separate wells of two 96-well microtiter plates. We used the same composition as in the glycerol media (minus the glycerol). Then, we grew each of the 31 strains in each of the 190 nutrients and negative controls (wells without carbon source) for 24 hr (shorter than 48 hr used previously to facilitate high throughput). We measured the final absorbance ($OD_{600}$) to quantify the ability to grow in that nutrient (*Figure 7A*; *Figure 7—figure supplement 1A*). The data showed that D-glucose (also known as dextrose) was the best carbon source for growth. Also, we saw 40 carbon

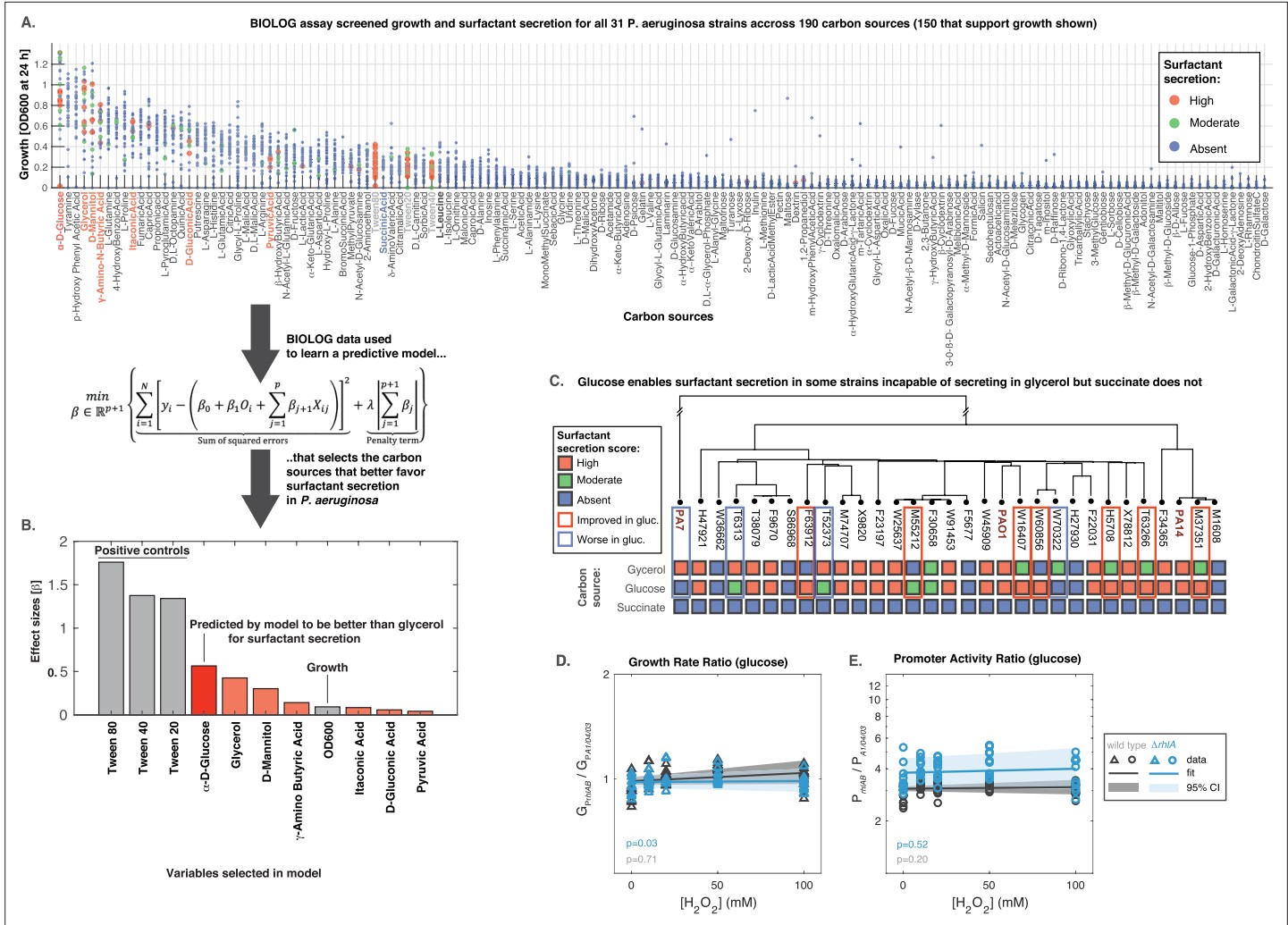

**Figure 7.** Glucose can make some strains secrete surfactants that could not do it from glycerol, further linking primary metabolism, oxidative stress, and surfactant secretion. (**A**) The 31 strains were profiled for growth and surfactant secretion in 190 carbon sources (see supporting **Figure 8** for full dataset), here ranked by their potential for growth (highest on top; the carbon sources selected by model highlighted in red; succinate in blue; and the tween positive controls in gray). (**B**) We trained a predictive model using the LASSO (least absolute shrinkage and selection operator), a supervised learning approach that shrinks linear regression parameters using an L1-penalty. The parameter fitting minimizes the sum of the squared errors of the model (left-hand term in the mathematical expression) and the sum of the absolute values of the betas, the model parameters (right-hand term in the mathematical expression) The values obtained for the parameters beta reveal D-glucose as the only carbon source (besides the positive controls) better than glycerol at inducing surfactant secretion. (**C**) Validation using glucose minimal medium reveals seven strains that improved surfactant secretion compared to glycerol, and four strains that worsened. This is in sharp contrast to succinate, a carbon source that imposes more oxidative stress and where no strain produces surfactants. (**D**) Similar experiment as in **Figure 6** but now adding $H_2O_2$ to PA14 growing in glucose. The extrinsic stress had no significant impact on the growth rate, even in the mutant lacking *rhlA*. (**E**) The *rhlAB* expression was also unaffected by $H_2O_2$, indicating that glucose places a lower burden on primary metabolism, enabling cells to both cope with oxidative stress and keep their level of the genes needed to make the surfactants.

The online version of this article includes the following figure supplement(s) for figure 7:

**Figure supplement 1.** D-glucose is predicted to induce surfactant secretion better than glycerol.

sources (including the non-metabolizable isomer of D-glucose, L-glucose) in which none of the strains could grow on ($OD_{600}$ <0.05).

Then we assessed the ability of each strain to secrete surfactants from each carbon source and gave a score of 0 (absent), 1 (moderate), or 2 (high) using the drop collapse assay. To avoid experimentalist bias, the drop collapse was done by an experimentalist blind to the surfactant secretion score that we had determined previously in glycerol. The final dataset totaled N=6014 data points (31

strains grown on 194 conditions each, including the 190 carbon sources and 4 negative controls)—a sizeable dataset that we used to parameterize the model.

As with any high throughput approach, the BIOLOG data came with a loss of precision. We could assess that imprecision first by looking at three positive controls—tween20, tween40, and tween80—which come in the PM1 BIOLOG plate and are themselves surfactants, which means that they should produce a score of 3 regardless of surfactant secretion. The error of failing to identify collapse in these compounds was 4.2%. Then, we collected additional BIOLOG data on the PA14 Δ*rhlA*, which cannot produce surfactants. The rate of false identification of collapse in this strain was 1.3%. Finally, we used the fact that the PM1 plates contain glycerol in one of the wells to compare those results with our previous surfactant secretion scores (*Figure 1*). The assay correctly determined 8/8 non-producers, which amount to a 100% specificity rate. It failed to detect 10/23 producers, which amount to a 43% sensitivity rate. Taken together, these results show that BIOLOG assay was mildly sensitive but highly specific: while it may miss some surfactant secretion cases, it is unlikely to overestimate the ability of a specific sugar to induce secretion.

In the model, we encoded each carbon source as a set of $p$ hot-encoded binary variables $X_i$. We included another covariate important to determine the surfactant production outcome: the $OD_{600}$ after at 24 hr (covariate $O_i$), which quantifies whether that strain could grow on that nutrient. The model therefore had $p + 1$ parameters, each represented by the Greek letter $\beta$ with an index ($i = 1, \ldots, p + 1$) where $p$ is the number of nutrient conditions tested ($\beta_1$ represents the parameter, or effect size, for the $OD_{600}$, and each $\beta_2$ to $\beta_{p+1}$ represents the parameters or effect sizes for each nutrient). We used the LASSO (*Tibshirani, 1996*) to impose an L1-penalization for the parameter $\lambda$ (see equation in *Figure 7*). Then we used threefold cross-validation, which splits the dataset into training and testing in repeated iterations and uses the test loss (the error rate of the model) to tune the penalty parameter. As common practice with LASSO, we choose the penalty value that was one standard error above the one that produced the minimum loss in the test: $\lambda = 9 \times 10^{-3}$ (*Figure 7—figure supplement 1B*). This penalty shrank all but 11 parameters to 0 (*Figure 7—figure supplement 1C*). The covariates with non-zero effect sizes included $OD_{600}$ and 10 carbon sources (*Figure 7D*). The top three compounds were—as expected—the positive controls tween80, tween40, and tween20. This provided a sanity check that the model worked as intended. D-glucose came in fourth place: the only carbon source that the model deemed better than glycerol at favoring *P. aeruginosa* surfactant production.

To validate the glucose result, we grew all the 31 strains in minimal medium using D-glucose as the sole carbon source, now using media that we made ourselves from scratch as opposed to the high throughput—but less precise—BIOLOG plates and we did each test in triplicate for added rigor. Consistent with our model, glucose improved surfactant secretion overall, showing better secretion seven strains relatively to the secretion we had seen in glycerol (*Figure 7C*); this included three strains that were non-producers in glycerol but turned into producers in glucose. Interestingly, the switch also went in the opposite direction: four strains had worse surfactant scores in D-glucose than in glycerol. This suggests a complex strain-to-strain variability in the metabolic features of *P. aeruginosa*. Nonetheless, the data showed that—at least for some strains—glucose, a carbon source which should impose less burden on primary metabolism (*Guerra-Santos et al., 1984*), could free up resources for secondary metabolism.

To test this idea further, we tested another carbon source, succinate, which *P. aeruginosa* can grow on (*Figure 7A*) but which—according to our model—should not improve secretion relatively to glycerol. Succinate enters the carbon metabolism directly through the TCA cycle, and compared to glycerol and glucose, the initial TCA cycle flux would be faster and generate more ROS level that would burden primary metabolism and impair surfactant synthesis. Indeed, none of the strains secreted surfactants in succinate (*Figure 7C*).

To investigate how oxidative stress might impact expression *rhlAB* in glucose we added $H_2O_2$ and monitored $P_{rhlAB}gfp$ fluorescence in the PA14 wild type and Δ*rhlA* strains. As described previously (*Figure 6*), we used the same constitutive $P_{A1/04/03}gfp$ as the baseline for comparison. Phase I growth rate was unaffected, indicating that both strains could still remove oxidative stress effectively (*Figure 7D*). As in glycerol, the *rhlAB* expression levels showed the same higher baseline level in the Δ*rhlA* compared to the wild type. But, contrasting with the results from glycerol, D-glucose did not increase *rhlAB* expression with increasing $H_2O_2$ in either strain (*Figure 7E*). Taken together, these

results support that carbon sources that place a lower burden on primary metabolism leave resources available to both deal with extrinsic oxidative stress and sustain surfactant production.

## Discussion

Here we investigated the evolution and regulation of surfactant secretion using diverse isolates of *P. aeruginosa*. Our results suggest a generalizable principle: that a strain can use its secondary metabolic pathways to produce surfactants if it can meet its primary requirements of energy production, biomass synthesis, and reduction of oxidative stress (*Figure 8A*). The lineages that retained their ability to make surfactants from glycerol were also those capable of reducing the oxidative stress created by primary metabolism and could overflow any excess carbon to the secondary metabolic pathway (*Figure 8B*). The inconsistent distribution across the phylogenetic tree was possibly due to the life histories of each lineage. Non-producing strains showed impaired abilities to reduce oxidative stress, a slower exponential growth in that carbon source (indicating a deficient primary metabolism), and perturbed levels of TCA cycle metabolites consistent with oxidative damage and accumulated intermediates of amino acid synthesis (also consistent with a slower primary metabolism). Those burdens may be what prevented the strains from making surfactants as their resources were prioritized for primary metabolism (*Figure 8C*). This was consistent with the observation the $OD_{600}$ at the end of exponential growth was not a good distinguisher for producers and non-producers, indicating that their final yield of biomass production was unaffected, while their exponential growth was.

What were the selective pressures that led to the diversity of surfactant secretion phenotypes? The answer to this question is less clear. The strains uncapable of making surfactants from glycerol-lacked genes (*Supplementary file 1*) that suggested a particular path which led to their loss of phenotype. Importantly, though, all three genes of the secondary biosynthetic pathway (*rhlA*, *rhlB*, *rhlC*) were intact, and some of those strains could make surfactants in D-glucose. Absence of a selective pressure to produce surfactants from glycerol in the life histories of non-producer lineages could account for these results. But another explanation is that the impaired ability to reduce oxidative stress in glycerol is a maladaptive consequence of specific adaptations experienced by those lineages. Since the strains were isolated from hospitalized patients—a host-associated environment where we may expect oxidative stresses imposed by the immune system in their fight against pathogens (*Zhu et al., 2019*) and by antibiotic treatment (*Mohamed et al., 2020*)—it is unlikely that the selection was for a worse response to oxidative stress.

Experiments in PAO1 and clinical isolates had shown previously that *P. aeruginosa* decreases expression of the key quorum-sensing genes (*lasI*, *lasR*, *rhlI*, *rhlR*, *pqsA*, and *pqsR*) in response to oxidative stress caused by exogenous $H_2O_2$ (*Mohamed et al., 2020*). Considering that quorum sensing regulates many genes (*Whiteley et al., 1999*), shutting down surfactant biosynthesis could be part of a broader response to reduce oxidative stress and save resources for primary metabolism. For PA14, a strain capable of reducing the stress produced by glycerol metabolism, adding an extrinsic oxidative stress induced expression of *rhlAB* in glycerol, though not in glucose. Δ*rhlA* increased the baseline expression in both glycerol and glucose but did not impact the rate of *rhlAB* increase with $H_2O_2$, suggesting an intricate regulation. Also, the lack of *rhlA* did not impact growth—in glycerol or glucose—compared to the wild type, even in the highest $H_2O_2$ concentrations tested. This is consistent with our flux-balance modeling of *Pseudomonas* metabolism, where shutting down rhamnolipid secretion did not impact the growth rate because the excess carbon could be alternatively secreted in other forms such as acetate (*Figure 4A*).

We did detect nuanced consequences of deleting surfactant biosynthesis in PA14: previous data shows that the Δ*rhlA* isogenic mutant of PA14 has lower levels of fMet (*Boyle et al., 2017*) which, interestingly, also happened in non-producer isolates analyzed here. fMet plays a role in translation initiation, and in the quality control mechanisms that degrade misfolded proteins (*Piatkov et al., 2015*). *Bacillus subtilis* lacking formyl-methionine transferase—the enzyme attaching a formyl group to methionine loaded on tRNA^fmet—is more sensitive to $H_2O_2$ and defective in swarming (*Cai et al., 2017*). This link between fMet, oxidative stress, and the loss of rhamnolipid synthesis is intriguing and warrants further research. Also, the Δ*rhlA* mutant had higher levels of gamma-glutamylcysteine (*Figure 3—figure supplement 4*), the immediate precursor of glutathione. Since glutathione is a well-known antioxidant that protects cells from oxidative damage (*Ezraty et al., 2017*), the result could indicate a mild oxidative stress. The finding that Δ*rhlA* expressed *rhlAB* at a higher level than

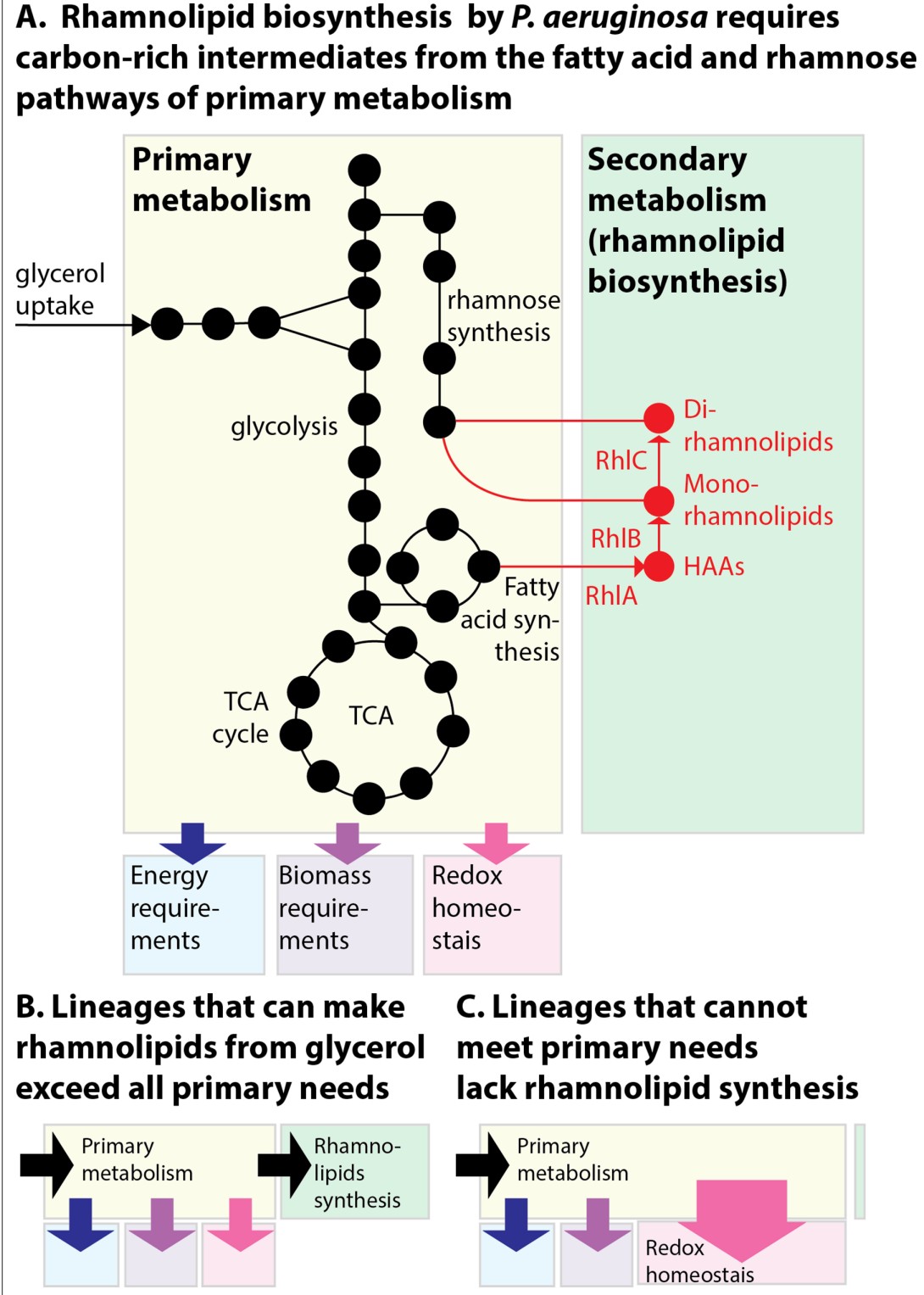

**Figure 8.** The allocation of metabolic resources made from glycerol in primary vs secondary metabolism. (**A**) Growth and rhamnolipid biosynthesis on glycerol as the sole carbon source place a strong burden on *Pseudomonas aeruginosa*: The bacterial cell has to make all the molecules needed for energy, biomass, and redox homeostasis; surfactant biosynthesis—a secondary metabolic pathway—competes for those resources. (**B**) *P. aeruginosa* lineages that retain the ability to make surfactants from glycerol meet their primary needs and use excess resources for secondary metabolism. (**C**) *P. aeruginosa* uncapable of making surfactants from glycerol was also worse at reducing the oxidative stress produced from growth in glycerol; the needs imposed on primary metabolism, such as maintaining redox homeostasis, may leave insufficient resources for secondary metabolism, explaining the loss of surfactant secretion.

the wild type—in glycerol and in glucose—suggests a futile attempt by this mutant to synthesize more surfactants. The fatty acid biosynthesis pathway that provides precursors for surfactant biosynthesis regenerates NAD(P)+ from NAD(P)H. $\Delta rhlA$ could perturb this pathway causing minor increases in oxidative stress.

Our analysis adds to the model of metabolic prudence, which provided a social evolutionary explanation for biosurfactant secretion. The model of metabolic prudence says that the microbial cell can lower the cost of a cooperative public good by delaying expression to times when it has excess carbon source (*Xavier et al., 2011*). A detailed study suggested that the strain PA14, needs to regulate *rhlAB* expression to carefully split the flux of carbon between biomass production and rhamnolipid synthesis (*Boyle et al., 2015*). Here, comparing diverse strains reminds us that heterotrophs like *P. aeruginosa* use organic compounds like glycerol, glucose, and succinate as carbon source for biomass, but also as an energy source. That energy fuels essential processes, including the reduction of oxidative stress, which ends up competing with secondary metabolism. This adds molecular detail to complement the evolutionary explanation and advance our understanding of *P. aeruginosa* surfactants.

More generally, our analysis highlights the importance of balancing the individual costs and population benefits of secondary metabolism. There are many products of secondary metabolism that contribute to the outsize impact that microbes have on the macroscopic world. Products of one microbial species can impact other species—cooperatively or competitively—potentially amplifying microbial functions in diverse communities such as the human microbiome (*Xavier, 2011*). The surfactants of *P. aeruginosa* are only one instance of the multitude of microbial secondary metabolism (*Demain, 2014*). But their analysis suggests a general principle: natural selection will favor sophisticated regulation that conciliates the individual-level costs and the population-level benefits of secondary metabolism.

## Materials and methods

### Media

Glycerol synthetic media were prepared with 800 ml of Milipore water, 200 ml of 5× minimal salts buffer, 1 ml of 1 M magnesium sulphate, 0.1 ml of calcium sulphate with 0.5 gN/l nitrogen (ammonium sulphate), iron at 5 µM (iron III sulphate), and 3.0 gC/l glycerol, respectively. 5× stock minimal salts buffer was prepared with 12 g of Na$_2$HPO4 (anhydrous), 15 g of KH$_2$PO4 (anhydrous), and 2.5 g of NaCl into 1 l water.

### Growth curve assays

The clinical isolates were inoculated in 3 mL of lysogen broth (LB) and incubated at 37°C overnight with shaking at 250 rpm. 500 µL of cell culture was pelleted and washed three times with PBS. 0.0025 units were inoculated into 3.0 g of carbon/liter of medium in glycerol minimal medium (7.7 g/l of glycerol) in BD Falcon (BD Biosciences, San Jose, CA) 96-well flat-bottom plates, with 150 µL of suspension per well. The plate was incubated during 48 hr at 37°C in a Tecan Infinite M1000 or Tecan Infinite M1000 Pro plate reader (Männedorf, Switzerland), with an orbital shaking of 4 mm of amplitude. OD$_{600}$ was measured in 10 min intervals.

### Rhamnolipid production quantification

The production of rhamnolipids was assessed by drop collapse assay (*Jain et al., 1991*; *Chen et al., 2007*). We placed 50 µL of the culture's supernatant on a polystyrene surface (the lid of a 96-well plate). The presence of rhamnolipids decreases the surface tension of the liquid, making the drop collapse. We considered a strain rhamnolipid non-producer if the drop does not spread, a mild-producer if spreads, but not enough to pass over the lid's circle that corresponds to the wells, and producer if spreads over it.

### Pangenome analysis

The pangenomes of our clinical *P. aeruginosa* isolates were identified using the roary tool (*Page et al., 2015*) using the output of genome annotation from prokka (*Seemann, 2014*). The core genes need to have at least 95% identity using blastp and to be present in at least 99% of all genomes, and then were aligned with the MAFFT method (*Katoh et al., 2002*).

## Phylogenetic analysis

The phylogenetic tree of clinical *P. aeruginosa* strains was constructed from core genomes as previously described (*Yan et al., 2017*). Moran's I test was carried out using the ape package in R (*Paradis and Schliep, 2019*). The ancestral state of swarming and rhamnolipid production was reconstructed using corHMM package (*Beaulieu et al., 2013*).

## Growth curve analysis

We first normalized the growth curves conducted in different experimental batches (different microtiter plates) using the PA14 inoculated in the same plate as the reference for normalization. The plate-specific coefficients were obtained by solving a linear regression model 'log(OD of PA14)~Plate + Time' (the Wilkinson notation) using Matlab function fitlm, where Plate is plate ids and Time is time points (both are categorical variables). Then the growth curves in each plate were normalized by multiplying the adjustment coefficient of the plate. These normalized growth curves were then smoothed by Savitzky-Golay filter (sgolayfilt function in Matlab). The growth phases (phase I, II, III) were determined by the following rules: (phase I) first-order derivative >0 and second-order derivative >0; (phase II) first-order derivative >0 and second-order derivative <0; (phase III) first-order derivative <0. The mean and maximum specific growth rate for each growth phase were estimated from the best-fit model among three models of bacterial growth curves: (*Morens et al., 2004*) exponential model $y(t) = y_0 + y_1 e^{\mu t}$ ; (*Rowland et al., 2018*) Zwietering-Logistic model $y(t) = y_0 + (A - y_0)\left(1 + e^{\frac{4\mu}{A}(\lambda - t)+2}\right)^{-1}$ ; (*Henderson, 1999*) Zwietering-Gompertz model $y(t) = y_0 + (A - y_0) e^{-e^{\frac{\mu e}{A}(\lambda - t)+1}}$ , where $y_0$, $y_1$, $\mu$, $\lambda$, $A$ are the parameters to fit. The goodness of fit for each model was evaluated by the value of the coefficient of determination, $R^2 = 1 - \frac{\sum_{t=1}^{n}(o(t)-y(t))^2}{\sum_{t=1}^{n}(o(t)-\bar{o})^2}$ , where $o(t)$ is the growth curve data observed at time point t and $\bar{o} = \frac{1}{n}\sum_{t=1}^{n} o(t)$. Both non-negative factorization and random forest regression were performed in python using sklearn.decomposition. NMF and sklearn.ensemble.RandomForestClassifier, respectively.

## Metabolite extraction

All *P. aeruginosa* strains were grown until the end of exponential phase of growth in glycerol minimal medium. Bacteria was then loaded into 0.25-µm nylon membranes (Millipore) using vacuum, transferred to pre-warmed hard agar plates with the same medium composition and incubated at 37°C during 2.5 hr. The filters were then passed to 35-mm polystyrene dishes (Falcon) with 1 mL of 2:2:1 methanol:acetonitrile:$H_2O$ quenching buffer and incubated there during 15 min on dry ice. Cells were removed by scraping, and the lysate containing quenching buffer was transferred to 1.5 mL tubes and centrifuged at 16,000 rpm for 10 min at 4°C. Supernatant transferred to fresh tubes and stored at –80°C.

## Metabolomic data preprocessing

The extracts were profiled using LC-MS, identifying a total of 92 compounds (*Supplementary file 5*). Some compounds contained missing values. These missing values in metabolite abundance can be (*Morens et al., 2004*) truly missing; (*Rowland et al., 2018*) present in a sample but its level is below detection limit; (*Henderson, 1999*) present in a sample at a level above the detection limit but missing due to failure of algorithms in data processing. Here we assume that a metabolite with missing values in all three replicates is truly missing in the sample and removed from our analysis (*Supplementary file 5*). However, if the missing values were only found in one or two replicates, the missing values were imputed by the average of the non-missing values. After that imputation, all compounds remaining with missing values were removed (*Supplementary file 5*).

The peak areas were normalized using cross-contribution compensating multiple standard normalization (*Redestig et al., 2009*) with NormalizeMets R package (*De Livera et al., 2018*). This method relies on the use of multiple internal standards. Since LC-MS lacks such internal standards, we used instead a set of metabolites assumed to be constant across all the strains. They were selected with a Kuskal-Wallis test, adjusting the p-value with Benjamini-Hochberg method. The ones with a p-value

above 0.05 were considered constant (pyruvate, methylglyoxal, (S)–2-acetolactate, tyramine, D-glu-cose, (S)-lactate, N-acetyl-L-glutamate 5-semialdehyde, 4-aminobutyraldehyde, and glycine), there-fore after the normalization step they were removed (indicated in red, *Supplementary file 5A*). The processed area peaks for all metabolites are included in *Supplementary file 5*.

## Unsupervised analysis of metabolomic data

The hierarchical clustering of the normalized metabolomic data was performed using gplots R package (*Warnes et al., 2015*), with Euclidean distance and Ward's aggregation method. The clustering was performed with all metabolites, including 16 unidentified metabolites that we could not identify (indi-cated in black, *Figure 2—figure supplement 2*). We included these 16 unidentified compounds in the clustering analysis because they were detected in all strains and therefore were not artifacts of the LC-MS. Nonetheless, because they were unidentified, we removed them from *Figure 3—figure supplement 2* and from the downstream analyses. Two other compounds, fumarate and guanosine, which were only putative initially, were bioinformatically confirmed as all other compounds with the same molecular weight are produced by enzymatic reactions missing in our clinical isolates. The PCA was done with R's base method, using also the normalized data and previously removing the 16 metabolites with unknown identity. The plot was obtained using ggplot2 R package (*Wickham, 2016*).

## Metabolic pathway enrichment

The differential metabolites between rhamnolipid producers and non-producers were determined by a Mann-Whitney test, with p-values adjusted with Benjamini-Hochberg method and a significance level of 0.05. These compounds were fed to FELLA algorithm (*Picart-Armada et al., 2018*; *Picart-Armada et al., 2017*). FELLA retrieves a graph describing the relationships among pathway, module, enzyme, reaction, and compounds of *P. aeruginosa* strain PA14 from the KEGG database. The graph was then used as an input network of differential compounds for its diffusion algorithms (*Vandin et al., 2011*). The output of FELLA consists of all subnetworks of the entries predicted to have a high connectivity with the differential compounds. We filtered the subnetwork entries to keep only meta-bolic pathways. The entries shown in *Supplementary file 3* are the ones with a significant probability of receiving part of the simulated flux.

## OPLS-DA model

OPLS-DA model of metabolomics data was built using ropls R package (*Thévenot et al., 2015*), fixing the number of orthogonal components to 3. $R^2$ and $Q^2$, key parameters for assessing the validity of the model, were assessed with sevenfold cross-validation. The significance of the model was determined by permutation test (n=2000). The p-value corresponds to the proportion of $Q^2_{perm}$ above $Q^2$. With a p-value below 0.05 we considered the model significant. The loadings of the predictive component of the model were extracted to determine how each metabolite contributes to the separation according to the phenotype.

## Genome-scale modeling

Custom Python codes were developed with the COBRApy package (*Ebrahim et al., 2013*) to carry out all metabolic flux modeling and simulations in the paper. Since iJN1411 model was developed for *P. putida*, we removed genes and associated reactions that are missing in all our strains but present in the iJN1411 model. The futile cycles involving NADH, NADPH, and GSH were also removed. The modified iJN1411 model was further expanded by adding rhamnolipid biosynthesis pathway involving 9 new metabolites and 12 new reactions.

The boundary fluxes of the model were set to mimic the composition of the glycerol minimum medium. For C:N=10, the lower bounds of glycerol and ammonium fluxes were set to –10 and –3, respectively. For C:N=3, the lower bounds were set to –3 and –3, respectively. The flux unit is mmol/gDW/hr throughout the paper. To constrain the total producing flux of NADH (the same for NADPH and GSH) at a certain value $C$, we first defined a binary variable $X_k$ for each NADH-involving reac-tion $k$ to indicate whether NADH is produced by this reaction. Given the stoichiometric coefficient of NADH in this reaction ($s_k$) and its flux value ($f_k$), the mathematical constraints for $X_k$ was set by $X_k = 1$ for $s_k f_k > 0$ and $X_k = 0$ otherwise. Therefore, the constraint that equalizes the total NADH producing flux and a constant $C$ is simply $\sum_k s_k X_k f_k = C$ . However, both $X_k$ and $f_k$ are variables and

such quadratic constraint has not yet been supported by COBRApy. We overcame this difficulty by defining $A_k = X_k f_k$ and linearized the product with the following two inequalities: $X_k l_b \leq A_k \leq X_k u_b$ and $f_k - (1 - X_k) u_b \leq A_k \leq f_k - (1 - X_k) l_b$ , where $l_b$ and $u_b$ are the lower and upper bounds of $f_k$ . The two constraints ensure that $A_k = 0$ when $X_k = 0$ and $A_k = f_k$ when $X_k = 1$. The minimum/maximum flux values of byproduct secretion were simulated by flux variability analysis at maximum growth rate.

## Detection of hydrogen peroxide (H₂O₂)

The $H_2O_2$ level in the extracellular medium was quantified with Amplex Red Hydrogen Peroxide/Peroxidase Assay Kit (Invitrogen, Carlsbad, USA, Catalog no. A22188). 500 µL of overnight growth in LB medium cell suspension were spun down, washed twice in PBS, and resuspended into glycerol medium at OD 1. Each reaction was done in a final volume of 100 µL, with a final concentration of 50 µM of Amplex Red reagent, 0.1 U/mL of HRP (Horseradish Peroxidase), and 0.2 cell OD, in glycerol synthetic medium, in BD Falcon (BD Biosciences, San Jose, CA) 96-well flat-bottom plates. The first column of the plate corresponded to the reaction without cells (the volume was substituted by PBS), and the last column instead of cells contained $H_2O_2$ (final concentration 10 µM). $OD_{600}$ and fluorescence 530/590 nm were measured in 10 min intervals (48 hr 37°C).

## RNA-seq and data analysis

The *P. aeruginosa* strains were inoculated into LB and grown with agitation for 16 hr. The overnight culture was diluted in glycerol synthetic medium the next day and grew to late exponential phase to harvest RNA. Cells were harvested by centrifuge and then immediately resuspended in RNAprotect (QIAGEN). The RNA was extracted using the RNeasy Mini Kit from QIAGEN. The standard RNA sequencing was done by Genewiz (Genewiz, Inc; South Plainfield, NJ) using pair-end indices of 150 bp read length on an Illumina HiSeq platform. The raw fastq files first went through quality check by fastQC (*Andrews, 2010*), and aligned with STAR (*Dobin et al., 2013*). The final output was generated using DEseq2 (*Love et al., 2014*) package in R. The read counts for each gene are included in *Supplementary file 6*. The SRA accession number for the RNAseq data is listed in *Supplementary file 7*.

## RLQ analysis

To test for the functional associations between genes and rhamnolipid production, we adopted the RLQ analysis (ade4 package in R) that was developed for identifying the associations between traits and environmental variables. Here we applied it to RNAseq analysis to understand the main co-structures between gene functions and phenotypes mediated gene expression. The theory of RLQ analysis is described in detail elsewhere (*Thioulouse et al., 2018*). Briefly, it builds on the simultaneous ordination of three tables: an R table (categorical) describing rhamnolipid production (strong-, mild-, and non-producers) for 11 selected isolates (each replicate was treated as an independent observation), an L table (continuous) describing the RNAseq counts of 1474 core genes shared among the 11 isolates, and a Q table (binary) describing 120 KEGG pathways for the 1474 genes. We first applied corresponding analysis (CA) to table L and Q and PCA to table R. In the CA of R and Q, each strain and gene were weighted by their total RNA expression levels, respectively. The three separate ordinations are then used as inputs to the rlq function. The output of the function includes coefficients for the gene functions and phenotypes (loadings) as well as the scores of strains (out of phenotypes) and scores of genes (out of gene functions). RLQ analysis maximizes the squared cross-covariances between the two sets of scores weighted by gene expressions.

## Hydrogen peroxide (H₂O₂) degradation

$H_2O_2$ was removed from environment by cells which degrade $H_2O_2$ intracellularly. The cumulative $H_2O_2$ removal curve was calculated by subtracting the values of emission of the wells containing each strain from the values of emission of the wells without cells or $H_2O_2$. Then they were normalized to the relative scale by dividing their values by the corresponding values (at the same time points) of the control strain (PA14) in the same experiment (plate). These normalized curves were then smoothed by the Savitzky-Golay filter (sgolayfilt function in Matlab). The $H_2O_2$ removal rate per cell was calculated by dividing the first-order derivative of the cumulative removal curves by OD values. To estimate the effect of rhamnolipid production on per-cell $H_2O_2$ removal rate, we developed a linear mixed-effect model that extends the simple linear regression model by considering both fixed and random effects.

The model was specified by the Wilkinson notation 'RemovalRate ~Time + RHL + (1|CurveID)', where RemovalRate is per-cell $H_2O_2$ removal rate, Time is time point (categorical variable), RHL is categorical rhamnolipid production (non-producer, mild-producer, and strong-producer), and CurveID is the unique ID for each strain and replicate. The expression '(1|CurveID)' indicates that CurveID was modeled as random effects. The model was solved by fitlme in Matlab.

## Promoter activity and growth rate comparison

Two *P. aeruginosa* strains carrying chromosomal insertions of the reporter fusion P*rhlAB*gfp (*Lequette and Greenberg, 2005*) and P*A1/04/03*gfp (a constitutive promoter, to be used as internal standard) (*Lambertsen et al., 2004*) in a PA14 background (*Xavier et al., 2011*), respectively, were inoculated into LB, and overnight cultures were harvested and washed with PBS three times. Cells were diluted to $OD_{600}$=0.1 in synthetic media and monitored using TECAN plate reader for both $OD_{600}$ and GFP fluorscence. Upon cells entering the exponential phase, $H_2O_2$ was added to the cells at a concentration in the assay of 0, 10, 20, 50, and 100 mM, continuing growth monitoring over 48 hr. After identifying growth phase II, the promoter activity at each timepoint within this interval was calculated as: $P = \frac{dGFP}{dt} \cdot \frac{1}{OD}$ . The ratio of promoter activity in each experiment was extracted as $R_p = \frac{median(PrhlAB)}{median(PA1/04/03)}$ . Similarly, the growth rate was calculated as $G = \frac{dOD}{dt} \cdot \frac{1}{OD}$ , and the ratio of strains with different promoters was calculated as $R_G = \frac{median(G_{PrhlAB})}{median(G_{PA1/04/03})}$ . The effect of $H_2O_2$ was estimated by fitglme in Matlab as: $\log(R) \sim a_0 + \beta \cdot [H_2O_2] + \epsilon$, where R is the measured ratio of either promoter activity or growth rate, and $\epsilon$ is denoted as the random effect of experiments. This procedure was used too for performing the promoter activity and growth rate analyses in PA14 Δ*rhlA* mutant.

## BIOLOG assay to obtain data for predictive modeling

We grew each of the *P. aeruginosa* stains in the BIOLOG phenotype microarrays PM1 and PM2a from Biolog Inc (Hayward, California, USA). We prepared minimal media without any carbon source added (800 ml of Milipore water, 200 ml of 5× minimal salts buffer, 1 ml of 1 M magnesium sulphate, 0.1 ml of calcium sulphate with 0.5 gN/l nitrogen [ammonium sulphate], iron at 5 µM [iron III sulphate]; 5× stock minimal salts buffer was prepared with 12 g of $Na_2HPO_4$ [anhydrous], 15 g of $KH_2PO_4$ [anhydrous], and 2.5 g of NaCl into 1 l water). We grow 5 mL bacterial cultures for 12–16 hr at 250 rpm and 37 °C, then aliquoted washed 1.5 mL in media without carbon source and adjusted the initial OD by further dilution. We added 150 ul of inoculated media onto each well of a BIOLOG plate and incubated for 24 hr overnight shaking at 250 rpm at 37°C. After that we measured the endpoint OD and assessed surfactant secretion using the drop collapse assay.

## Acknowledgements

This work was supported by NIH grants U01 AI124275 and R01 AI137269-01 to JBX, FRP and GS were partially supported by UIDB/04046/2020 and UIDP/04046/2020 Centre grants from FCT, Portugal (to BioISI) and by the LungCARD project (Grant Agreement n°: 734790, H2020-MSCA-RISE-2016). GS is the recipient of a fellowship from BioSys PhD programme PD65-2012 (Ref SFRH/BD/142899/2018) from FCT (Portugal). The funders had no role in study design, data collection and analysis, decision to publish, or preparation of the manuscript.

## Additional information

### Funding

| Funder | Grant reference number | Author |
|---|---|---|
| National Institutes of Health | U01 AI124275 | Joao B Xavier |
| National Institutes of Health | R01 AI137269 | Joao B Xavier |

| Funder | Grant reference number | Author |
| --- | --- | --- |
| Fundação para a Ciência e a Tecnologia | UIDB/04046/2020 | Francisco Rodrigues Pinto |
| Fundação para a Ciência e a Tecnologia | UIDP/04046/2020 | Francisco Rodrigues Pinto |
| European Research Council | 734790 | Francisco Rodrigues Pinto |
| Fundação para a Ciência e a Tecnologia | SFRH/BD/142899/2018 | Guillem Santamaria |

The funders had no role in study design, data collection and interpretation, or the decision to submit the work for publication.

## Author contributions

Guillem Santamaria, Conceptualization, Software; Chen Liao, Conceptualization, Resources, Data curation, Software, Validation, Visualization, Methodology, Writing – original draft, Writing – review and editing; Chloe Lindberg, Investigation, Writing – review and editing; Yanyan Chen, Formal analysis, Writing – review and editing; Zhe Wang, Investigation, Methodology; Kyu Rhee, Supervision, Methodology; Francisco Rodrigues Pinto, Conceptualization, Supervision; Jinyuan Yan, Conceptualization, Resources, Data curation, Software, Investigation, Methodology, Writing – original draft, Project administration, Writing – review and editing; Joao B Xavier, Conceptualization, Resources, Formal analysis, Supervision, Funding acquisition, Investigation, Methodology, Writing – original draft, Project administration, Writing – review and editing

## Author ORCIDs

Chen Liao (ID) http://orcid.org/0000-0001-8474-1196
Francisco Rodrigues Pinto (ID) http://orcid.org/0000-0002-4217-0054
Jinyuan Yan (ID) http://orcid.org/0000-0003-2046-5625
Joao B Xavier (ID) http://orcid.org/0000-0003-3592-1689

## Decision letter and Author response

Decision letter https://doi.org/10.7554/eLife.76119.sa1
Author response https://doi.org/10.7554/eLife.76119.sa2

# Additional files

## Supplementary files

• Supplementary file 1. Presence and absence of genes across the genomes of our clinical isolates.

• Supplementary file 2. Quantitative values for the seven local features of phase I, II, and III (phase start time point, phase duration, phase initial OD, OD change, area under the curve, mean specific growth rate, and maximum specific growth rate) of growth curves.

• Supplementary file 3. Different pathways that are significantly different in rhamnolipid producers and non-producers identified by FELLA.

• Supplementary file 4. Correlations between gene expressions or pathways with rhamnolipid production in RLQ analysis. For each rhamnolipid production category (strong-, mild-, and non-producers), its correlation value with a single gene or a functional pathway was computed as the dot products between the arrow of phenotypic category and the arrow of the gene or pathway in RLQ axes.

• Supplementary file 5. Normalized peak area of metabolomics.

• Supplementary file 6. Read counts of *P. aeruginosa* RNAseq aligned to the PA14 genome.

• Supplementary file 7. Metadata of RNAseq archived in NCBI SRA database.

• Transparent reporting form

## Data availability

Sequencing data have been deposited in SRA, in the bioproject accession number PRJNA253624. Each individual sample has a file accession number listed in Supplementary file 7. The additional dataset is

provided through Dryad. The source code for generating the figures of this study is available from: https://github.com/guisantagui/code_PA_paper (copy archived at swh:1:rev:cf190909e0f87db785ec4803bd-d6e1a7cb854ad9); https://github.com/liaochen1988/Source_code_for_Pseudomonas_Metab-olomics_Paper (copy archived at swh:1:rev:dcc50627d31d66a9f25a4d6b60294321087edf50); https://github.com/Jinyuan1998/PA_metabolomics_rhamnolipids_SourceCode (copy archived at swh:1:rev:9c8f53d6d21bd089c605f8b55f72f04774fd31bd).

The following datasets were generated:

| Author(s) | Year | Dataset title | Dataset URL | Database and Identifier |
|---|---|---|---|---|
| Yan J | 2021 | RNAseq files | https://www.ncbi.nlm.nih.gov/bioproject/?term=PRJNA253624 | NCBI BioProject, PRJNA253624 |
| Xavier J, Pinto FR, Wang Z, Rhee K, Liao C, Santamaria G, Yan J | 2021 | Evolution and regulation of microbial secondary metabolism | https://dx.doi.org/10.5061/dryad.7sqv9s4tg | Dryad Digital Repository, 10.5061/dryad.7sqv9s4tg |

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
