## [Editor Report]

This important study proposes how *Pseudomonas aeruginosa* is able to produce costly secondary metabolites. By combining a wide variety of experimental and computational approaches and studying 31 isolates in several growth environments, they present highly compelling evidence that secondary metabolites are produced in this species when it is not suffering from oxidative stress linked to its primary metabolism. This paper helps to understand pathogen metabolism, and more generally, how environmental conditions can allow for the evolution of costly collective behavior.

---

## [Decision Letter]

**Decision letter after peer review:**

Thank you for submitting your article "Evolution and regulation of microbial secondary metabolism" for consideration by *eLife*. Your article has been reviewed by 2 peer reviewers, one of whom is a member of our Board of Reviewing Editors, and the evaluation has been overseen by Gisela Storz as the Senior Editor. The reviewers have opted to remain anonymous. We would like to apologize for the delay in reviewing your paper. Finding reviewers over the holiday period and in the weeks after was quite challenging.

Essential revisions:

Both reviewers found the depth of the analysis to be highly impressive and we would like to congratulate you on a beautiful piece of work! The main criticism by both reviewers was the relevance of using glycerol as a sole carbon source is for clinical *Pseudomonas aeruginosa* strains. One way to address this would be to use the model to predict what environment would lead the non-producers to produce rhamnolipids and test it experimentally. Such additional data would help in extrapolating to the natural environment, and would help to justify the use of glucose. We realize that this may be quite a lot of additional work, but I am afraid that without it, the arguments presented around the relevance of the growth condition would need to be much stronger for the paper to be accepted with the current data, despite its depth and rigor. We also found that the readability of the paper and the way the story is presented could be improved.

*Reviewer #1 (Recommendations for the authors):*

The one set of experiments that would be great to see is if one could reduce oxidative stress for non-producers and to see whether they would then produce rhamnolipids. If I understand correctly, that would be the hypothesis from this work? It seems to me that this would not be too much additional work, particularly since the metabolic model could help predict the right conditions. Or one could at least use a few representative strains and conduct a quick screen of different carbon sources. Or perhaps it is the C:N ratio that needs to be changed?

Some more detailed comments:

– L. 44: I would argue r is typically high in microbial populations due to clonal cell division. In any case, this is not very important for the argument of the paper overall. Also, you resolve it around l. 76, which makes me wonder whether it was necessary to discuss.

– How do you know that non-producers are not just in the wrong conditions (e.g. carbon to nitrogen ratio).

– L. 100: what is a "drop collapse assay"?

– L. 101 why "surface tension"? How does that indicate surfactant activity?

– L. 159-160: which metabolites? Same as the 15 in l. 161?

– L. 176ff: might be worth expanding a bit more on the DrhlA mutant and how it relates to all of the non-producing strains.

– L. 399: gC/l?

– L. 418ff: I did not understand the normalisation.

– L. 427: can you add more detail on how you decide on the best fit? R2 of what and using what test?

– L. 466: if unknown compounds were not artefacts, why were they removed?

– L. 483: What is OPLS-DA?

– L. 512ff: you grow cells in LB for the H2O2 assays? Shouldn't it be the glycerol medium?

– L. 564ff: Where did the GFP strains come from?

– Figure 3D: OD_600?

– Figure 4A, highlighting in black does not really help to see the arrows, it might also be useful to put a table that figure with the pathways, rather than in the caption.

– Supp. Figure 2: what does presence/absence of the accessory genomes mean?

– Supp. Figure 4: it would be good to highlight which ones are rhamnolipid+ or -.

– Supp. Figure 8: which one is higher? What are blue/red dots?

[Editors’ note: further revisions were suggested prior to acceptance, as described below.]

Thank you for resubmitting your work entitled "Evolution and regulation of microbial secondary metabolism" for further consideration by *eLife*. Your revised article has been evaluated by Naama Barkai (Senior Editor) and a Reviewing Editor.

The manuscript has been improved but there are some remaining issues that need to be addressed, as outlined below:

As reviewer #2 has detailed, the introduction and discussion need to be better written to make the article more accessible. This would greatly increase the impact of this very nice work.

*Reviewer #1 (Recommendations for the authors):*

The authors have done an impressive amount of additional work to show that the principles they uncovered initially also hold for different carbon sources. This again makes use of predictive modeling, showing that these results may be extrapolated for other environments that were not included in this study, assuming that similar in vitro experiments can be done to parameterize the model. Although this does not address the point of medical relevance, it does increase the generality of the finding, which I think greatly strengthens an already strong paper.

*Reviewer #2 (Recommendations for the authors):*

I was pleased to see that the authors made a great effort to address the most important comment during the first round of review, i.e. the relevance of growth on glycerol as the sole carbon source. The authors performed an extensive biolog screening and modeling that showed that carbon sources that put little strain on the metabolism allow surfactant production, while carbon sources that put more strain on the primary metabolism do not allow surfactant production. This contributes to the generalizability of the main conclusion that '*P. aeruginosa* lineages produce surfactants if they can reduce the oxidative stress produced by primary metabolism and have carbon source in excess of their growth requirements'.

However, during the first review round both reviewers also made comments about the readability of the text and the build-up of the story. I have to say that I found the revised text still very complex and difficult to read. The text starts with a broad introduction on social evolution and the paradox of why cells invest into rhamnolipid production, while the emphasis in the Results section is rather on understanding *P. aeruginosa* metabolism. Also, the discussion does not extensively or explicitly return to the relevance of the results for social evolution theory. I regret that the authors did not implement or answer my earlier suggestion to 'systematically compare similarities and differences between the proposed mechanism and earlier described mechanisms of metabolic prudence'. Several mechanisms of metabolic prudence have been described. Making this comparison would allow to explicitly return to the concepts of social evolution theory highlighted in the introduction, and as such both increase the readability and provide a more nuanced view of the novelty of the proposed mechanism. Alternatively, the introduction section can be simplified by removing the social evolution concepts and introducing them only later on in the discussion.

---

## [Author Response]

Essential revisions:Both reviewers found the depth of the analysis to be highly impressive and we would like to congratulate you on a beautiful piece of work! The main criticism by both reviewers was the relevance of using glycerol as a sole carbon source is for clinical *Pseudomonas aeruginosa* strains. One way to address this would be to use the model to predict what environment would lead the non-producers to produce rhamnolipids and test it experimentally. Such additional data would help in extrapolating to the natural environment, and would help to justify the use of glucose. We realize that this may be quite a lot of additional work, but I am afraid that without it, the arguments presented around the relevance of the growth condition would need to be much stronger for the paper to be accepted with the current data, despite its depth and rigor. We also found that the readability of the paper and the way the story is presented could be improved.

We are glad that the referees agreed favorably. In this revised version we addressed the essential issue raised by everyone on the relevance of glycerol. This was indeed the only carbon source that we described in our original submission. We have now collected additional data and we wrote a new section at the end of the Results section where we:

– Compared a wide range of carbon for the ability to grow *P. aeruginosa* and secrete surfactants. We do this using the BIOLOG PM1 and PM2a carbon source arrays.

– Developed a mathematical model to predict whether a strain produces surfactants from a specific carbon source, and we used the model to identify D-glucose as a preferred carbon source—better than glycerol— to induce surfactant secretion. We also tested succinate as a carbon source with the opposing effect: none of the strains produces surfactants despite being able to grow well on this carbon source.

– Combined the new experimental and modelling results to refine our understanding of the process. The refined theory explains that each carbon source imposes a different burden on primary metabolism. The new results provide further support to the notion that surfactant secretion is highest in the carbon sources that impose low levels oxidative stress on primary metabolism, such as D-glucose, and lowest in carbon sources that impose stress such as succinate.

We believe this new section improved our paper by adding to our understanding of how primary metabolism conditions biosurfactant secretion in *P. aeruginosa*.

Reviewer #1 (Recommendations for the authors):The one set of experiments that would be great to see is if one could reduce oxidative stress for non-producers and to see whether they would then produce rhamnolipids. If I understand correctly, that would be the hypothesis from this work? It seems to me that this would not be too much additional work, particularly since the metabolic model could help predict the right conditions. Or one could at least use a few representative strains and conduct a quick screen of different carbon sources. Or perhaps it is the C:N ratio that needs to be changed?

We thank the referee for this recommendation. This let us to add the new section that we mentioned above. We acquired new data and refined our model beyond glycerol. The new data supports that the surfactant secretion happens in carbon sources that impose low oxidative stress.

Some more detailed comments:– L. 44: I would argue r is typically high in microbial populations due to clonal cell division. In any case, this is not very important for the argument of the paper overall. Also, you resolve it around l. 76, which makes me wonder whether it was necessary to discuss.

We have revised the section mentioned by the referee. We now acknowledge that clonal division should increase the *r* (p. 3, l. 47). We also edited the introduction substantially, and we hope to have made it clearer. The new introduction now stresses better the previous research was all done strains, which leaves open the question of why some strains can produce rhamnolipids when others fail to do so.

– How do you know that non-producers are not just in the wrong conditions (e.g. carbon to nitrogen ratio).

This is an important question that we hope to have addressed in the revised version. Our new analysis of surfactant biosynthesis in a range of carbon sources refined our theory. We now understand better that surfactants results from a combination of high carbon and a low oxidative stress. What we called nonproducers before are in fact strains with a lower tolerance to the oxidative stress produced in glycerol. Carbon sources that impose a higher stress (e.g. succinate) can make all strains behave as non-producers.

– L. 100: what is a "drop collapse assay"?

We added new panels to Figure 1 (Figures 1A and 1B) to explain the assay.

– L. 101 why "surface tension"? How does that indicate surfactant activity?

The activity of the biosurfactant is to reduce the surface tension of the surrounding liquid. We hope the revision in the previous comment makes this clearer for the reader.

– L. 159-160: which metabolites? Same as the 15 in l. 161?

The OPLS-DA searched for broad associations between the general metabolome and a dependent variable, in this case the surfactant secretion. We revised that paragraph hoping to clarify the point for the readers. WE now say:

“Fitting the data using the Orthogonal Projections to Latent Structures-Discriminant Analysis (OPLSDA) (48) confirmed that there is a significant association between the broad metabolome and surfactant secretion (Figure 3A, R^2^ = 0.82, Q^2^ = 0.66, *p-*value = 5e-4). A Mann-Whitney U test revealed the 15 metabolites whose abundances differed most significantly between producers and non-producers. Then, we used those 15 metabolites to identify pathways perturbed in non-producers using the FELLA algorithm for pathway enrichment analysis (49) (Supplementary Figure 7, Supplementary Table 3).”

– L. 176ff: might be worth expanding a bit more on the DrhlA mutant and how it relates to all of the non-producing strains.

We have expanded the entire paragraph to explain better the DrhlA mutant and how it relates to the nonproducing strains. The new paragraph says:

“To place this observation in context, we reanalyzed previous metabolomics data (50) from an experiment were we had compared PA14 and its Δ*rhlA* mutant. Interestingly, the Δ*rhlA* mutant also had lower levels of fMet compared with the wild type (Supplementary Figure 8)*.* The Δ*rhlA* mutant grows just as fast in glycerol as the wild type (51), which means that the link between lower fMet and lack of surfactant secretion is not simply due to a growth defect.” (p. 9, l. 188-192)

– L. 399: gC/l?

We chose to provide the media in grams of carbon per liter of medium (gC/l). We believe that this fits better the present work, with its focus on the carbon source. But we can see that this choice might have been confusing. We have now extended the protocol description to explain this better:

“3.0 gram of Carbon/liter of medium in glycerol minimal medium (7.7 g/l of glycerol)” (p. 23, l. 511-512).

– L. 418ff: I did not understand the normalisation.

We rephrased the description of the normalization to explain better:

“We first normalized the growth curves conducted in different experimental batches (different microtiter plates) using the PA14 inoculated in the same plate as the reference for normalization” (p. 24, l. 531-534).

– L. 427: can you add more detail on how you decide on the best fit? R2 of what and using what test?

We now explain that R2 is the coefficient of determination. We provide the formula used to calculate the R2 (p. 24, l. 545).

– L. 466: if unknown compounds were not artefacts, why were they removed?

We see that our explanation was confusing. The 16 unidentified metabolites were included in the clustering analysis: they were only removed for downstream analysis where the lack of identity would prevent biological interpretation. We have edited the sentence noted by the referee to make it clearer:

“The clustering was performed with all metabolites, including 16 unidentified metabolites that we could not identify (indicated in black, Supplementary Figure 4). We included these 16 unidentified compounds in the clustering analysis because they were detected in all strains and therefore were not artifacts of the LC-MS. Nonetheless, because they were unidentified, we removed them from Supplementary Figure 6 and from the downstream analyses.” (p. 26, l. 580-584).

– L. 483: What is OPLS-DA?

We now explain that OPLS-DA stands for Orthogonal Projections to Latent Structures Discriminant Analysis (p. 27, l. 600).

– L. 512ff: you grow cells in LB for the H2O2 assays? Shouldn't it be the glycerol medium?

We apologize for the lack of clarity: The H2O2 assay was indeed in glycerol medium. It was the overnight culture that was done in LB. We edited the text to clarify this point:

“500 μL of overnight growth in LB medium cell suspension were spun down, washed twice in PBS and resuspended into glycerol medium at OD 1” (p. 28, l. 631).

– L. 564ff: Where did the GFP strains come from?

We expanded the protocol noted by the referee to explain that the *P. aeruginosa* strains carrying chromosomal insertions of the reporter fusion P*_rhlAB_gfp* in a PA14 background came from two previous studies. The P*_rhlAB_gfp* fusion was originally made for strain PAO1 by Lequette *et al.* (2005). It was put into PA14 by Xavier *et al.* (2011).

– Figure 3D: OD_600?

We now explain that OD_600_ means the population optical density at 600 nm (p. 6, l. 136). We also have edited all the figures showing OD_600_ data to explicitly say so.

– Figure 4A, highlighting in black does not really help to see the arrows, it might also be useful to put a table that figure with the pathways, rather than in the caption.

We redid the former Figure 4 (now Figure 5) completely to improve clarity:

– We added a new panel explaining the RLQ analysis.

– We used colors to highlight to highlight the genes and pathways associated with surfactant secretion phenotypes

– We added a panel with the pathways as requested by the referee.

– Supp. Figure 2: what does presence/absence of the accessory genomes mean?

We expanded the caption of Supp. Figure 2 to explain:

“A principal component analysis (PCA) of the matrix of presence/absence of genes in each strain. The matrix was constructed using the set of all gene in the accessory genome (genes present in some strains but not all) shows no obvious differences distinguishing surfactant producers from non-producers.”

– Supp. Figure 4: it would be good to highlight which ones are rhamnolipid+ or -.

We added a label to each plot to indicate the label of each strain.

– Supp. Figure 8: which one is higher? What are blue/red dots?

We expanded the caption of Supp. Figure 8 to explain:

“A positive log2 fold change indicates a metabolite enriched in the mutant compared to the wild type (WT). The dots in red represent metabolites significantly enriched in the mutant, the dots in blue represent metabolites significantly reduced in the mutant.”

[Editors’ note: further revisions were suggested prior to acceptance, as described below.]

The manuscript has been improved but there are some remaining issues that need to be addressed, as outlined below:As reviewer #2 has detailed, the introduction and discussion need to be better written to make the article more accessible. This would greatly increase the impact of this very nice work.

Thank you for the compliment on our work. We have revised the introduction and the discussion. We hope our changes have made the article more accessible.

Reviewer #1 (Recommendations for the authors):The authors have done an impressive amount of additional work to show that the principles they uncovered initially also hold for different carbon sources. This again makes use of predictive modeling, showing that these results may be extrapolated for other environments that were not included in this study, assuming that similar in vitro experiments can be done to parameterize the model. Although this does not address the point of medical relevance, it does increase the generality of the finding, which I think greatly strengthens an already strong paper.

We are pleased that the referee agrees that the data in different carbon sources has strengthened the paper. We thank the reviewer for their compliment of our paper, and for the constructive criticism that led to the improvements.

The referee notes that our changes do not address the point of medical relevance. We have improved the last sentence of the abstract in response to this. It now reads:

“These results add a new layer to the regulation of a metabolite unessential for the primary metabolism of the bacterium but important to change the physical properties of its surrounding environment.”

Reviewer #2 (Recommendations for the authors):I was pleased to see that the authors made a great effort to address the most important comment during the first round of review, i.e. the relevance of growth on glycerol as the sole carbon source. The authors performed an extensive biolog screening and modeling that showed that carbon sources that put little strain on the metabolism allow surfactant production, while carbon sources that put more strain on the primary metabolism do not allow surfactant production. This contributes to the generalizability of the main conclusion that '*P. aeruginosa* lineages produce surfactants if they can reduce the oxidative stress produced by primary metabolism and have carbon source in excess of their growth requirements'.However, during the first review round both reviewers also made comments about the readability of the text and the build-up of the story. I have to say that I found the revised text still very complex and difficult to read. The text starts with a broad introduction on social evolution and the paradox of why cells invest into rhamnolipid production, while the emphasis in the Results section is rather on understanding *P. aeruginosa* metabolism. Also, the discussion does not extensively or explicitly return to the relevance of the results for social evolution theory. I regret that the authors did not implement or answer my earlier suggestion to 'systematically compare similarities and differences between the proposed mechanism and earlier described mechanisms of metabolic prudence'. Several mechanisms of metabolic prudence have been described. Making this comparison would allow to explicitly return to the concepts of social evolution theory highlighted in the introduction, and as such both increase the readability and provide a more nuanced view of the novelty of the proposed mechanism. Alternatively, the introduction section can be simplified by removing the social evolution concepts and introducing them only later on in the discussion.

We have rewritten the introduction and the Discussion sections to improve clarity. We paid great attention to the referee’s request, especially their point on comparing with earlier work, both in the introduction and in the discussion, and we revisit the concept of metabolic prudence in the discussion to return to the social evolutionary concepts from the introduction. We hope that the rewrite has improved the paper.